# 3D-printed liquid metal polymer composites as NIR-responsive 4D printing soft robot

Liwen Zhang[1], Xumin Huang [1], Tim Cole [2], Hongda Lu [1,3], Jiangyu Hang[1], Weihua Li [3], Shi-Yang Tang [4] ✉, Cyrille Boyer [5], Thomas P. Davis [1] ✉ & Ruirui Qiao [1] ✉

4D printing combines 3D printing with nanomaterials to create shape-morphing materials that exhibit stimuli-responsive functionalities. In this study, reversible addition-fragmentation chain transfer polymerization agents grafted onto liquid metal nanoparticles are successfully employed in ultra-violet light-mediated stereolithographic 3D printing and near-infrared light-responsive 4D printing. Spherical liquid metal nanoparticles are directly prepared in 3D-printed resins via a one-pot approach, providing a simple and efficient strategy for fabricating liquid metal-polymer composites. Unlike rigid nanoparticles, the soft and liquid nature of nanoparticles reduces glass transition temperature, tensile stress, and modulus of 3D-printed materials. This approach enables the photothermal-induced 4D printing of composites, as demonstrated by the programmed shape memory of 3D-printed composites rapidly recovering to their original shape in 60 s under light irradiation. This work provides a perspective on the use of liquid metal-polymer composites in 4D printing, showcasing their potential for application in the field of soft robots.

4D printing represents a cutting-edge technology in additive manufacturing, wherein initially static 3D-printed materials undergo shape transformations over time[1,2]. 4D printing technology harnesses programmable and advanced smart materials that respond to specific stimuli, such as water[3], heat[4,5], photo[6], and pH[7], to achieve shape and property changes. This technology creates objects with customizable and controllable shape transformation by harnessing shape memory effects[8,9], opening up various avenues for soft robotics[10,11], wearable devices[12], and healthcare[13]. Conventional 3D-printed polymers display limited responses to external stimuli such as light, electricity, and magnetic fields, limiting their utility in 4D printing applications. To overcome the limitations of using polymers alone, several researchers have demonstrated the integration of functional rigid nanomaterials into 3D-printed presents several benefits for advancing the 4D printing

nanocomposites[14]. Nanoparticles with unique properties (such as photosensitivity and chemosensitivity) can enhance the stimuli-responsiveness of printed objects, thereby enabling precise and efficient shape changes. For instance, Kuhnt et al. successfully integrated magnetic iron oxide nanoparticles into 3D-printed objects, thereby producing composites that demonstrated thermally and remotely controlled shape-memory behavior under an alternating magnetic field[15]. An alternative approach to producing composites capable of inducing shape transformation involved the incorporation of conductive carbon black particles or carbon nanotubes into 3D-printed polymers, thus sensitizing the composites to electrothermal stimulation. Moreover, rigid nanomaterials can also reinforce the mechanical properties of 4D-printed products, elevating overall durability and augmenting the load-bearing capacity of the composite material for diverse applications.

[1]Australian Institute of Bioengineering & Nanotechnology, The University of Queensland, Brisbane, QLD 4072, Australia. [2]Department of Electronic, Electrical, and Systems Engineering, University of Birmingham, Birmingham, UK. [3]School of Mechanical, Materials, Mechatronic and Biomedical Engineering, University of Wollongong, Wollongong, NSW 2522, Australia. [4]School of Electronics and Computer Science, University of Southampton, Southampton SO17 1BJ, UK. [5]Cluster for Advanced Macromolecular Design, School of Chemical Engineering, The University of New South Wales, Sydney, NSW 2052, Australia. ✉e-mail: shiyang.tang@soton.ac.uk; t.davis@uq.edu.au; r.qiao@uq.edu.au

Nevertheless, a drawback to using particulate-based nanocomposites to achieve shape change lies in establishing the necessary percolated network within the elastomer matrix, often requiring high filler loadings. These elevated loadings complicate the printing process and limit the extent of shape change[16,17]. Furthermore, the utilization of rigid nanoparticles may compromise the softness and flexibility of composites, thereby limiting the shape-morphing capabilities of 4D printed materials in intricate environments. In addition, some studies have reported that integrating rigid nanoparticles can increase the glass transition temperature ($T_g$) and melting temperature ($T_m$) of polymers in thermal-responsive 4D printing[18,19], leading to an increase in the temperature required for shape recovery, and constraining the spectrum of stimuli capable of inducing deformation.

Gallium-based liquid metals (LMs), as metallic fluids at room temperature[20–22], have attracted considerable interest for stretchable devices[23], and soft robotics[24]. In addition to the favorable thermal and electrical properties of metallic materials[25], LMs offer unique advantages in terms of fluidity, deformability, low melting points, and self-healing abilities[26–29]. In recent work, LM-polymer composites have been reported to exhibit thermally or electrically induced shape memory characteristics through the integration of LMs within polymeric materials[30,31]. In contrast with hard materials, the unique fluidic nature of LMs prevents dramatic changes of the mechanical properties and allows for the deformation of LMs within the polymer matrix to further improve shape recovery behavior[32]. While 4D printing with LMs shows promise, it does face certain challenges. These include the inevitable aggregation and precipitation of LMs during the printing process, as well as their susceptibility to oxidation, which can alter the physical and chemical properties of the printed materials. Overcoming these limitations will be crucial in broadening the application of LMs in 4D printing and shape memory polymers.

Herein, we report an innovative synthetic approach for the fabrication of 4D printing LM/polymer composites with shape memory properties under near-infrared NIR irradiation. This was achieved through the UV-mediated 3D printing of liquid metal nanoparticles (LMNPs) grafted with reversible addition–fragmentation chain-transfer (RAFT) agents to enable polymerization for 3D printing. The RAFT agent grafted LMNPs as RLMNPs exhibited a notably increased surface area-to-volume ratio and reduced diameters, which significantly enhanced their stability and dispersity in various solvents when compared to bulk gallium-indium eutectic (EGaIn). Surface grafting with functional ligands has been demonstrated as an effective approach to improve the dispersity of LMNPs in solutions[33,34], and prevent surface oxidation by creating a protective barrier between the metal and the surrounding environment[35,36]. Most importantly, RLMNPs can be directly prepared in 3D-printed resins, providing a simple and effective one-step printing approach for manufacturing liquid metal polymer composites (LMPCs). The RLMNPs were successfully employed in stereolithographic 3D printing to fabricate LMPCs via type I photoinitiated RAFT polymerization. The RLMNPs presented uniform distribution in 3D-printed polymer composites. With the integration of RLMNPs, our 3D-printed LMPCs exhibited excellent mechanical properties including reduced $T_g$, storage modulus (G′), tensile stress (σ), and Young's modulus (E) of 3D-printed objects. These composites exhibited notable efficiency in light-triggered 4D printing, rapidly and completely restoring their pre-programmed shape when exposed to NIR light irradiation for 60 s. The process of deformation and recovery was repeated without any noticeable decline in recovery efficiency and mechanical properties. The shape memory feature of the composites allowed them to lift loads to five times their weight. Furthermore, 3D-printed LMPCs serve as NIR-responsive soft robot that extends beyond mere grasping and releasing items, encompassing the additional capability to control rotational bistable structures. The exceptional features of RLMNPs-based 4D printing make it a highly promising technology for developing the next generation of soft robotics, artificial muscles, and aerospace engineering materials.

## Results and discussion

### Preparation and characterization of liquid metal nanoparticles

We prepared the RLMNPs by using a diphosphonic acid group terminated RAFT agent (Fig. 1a) through the strong coordination binding between Ga and phosphonic acid[33,37]. 2-(((butylthio)carbonothioyl)thio)propanoic acid terminated with diphosphonic acid (as RAFT agents) and EGaIn were simultaneously sonicated in the ethanol to obtain RLMNPs (Fig. 1a). Transmission electron microscopy (TEM) images revealed that the RLMNPs exhibited a spherical morphology, with an average diameter of approximately 204 nm (Fig. 1b), which was larger than the LMNPs with a diameter of about 107 nm (Supplementary Fig. 1a). The larger size of RLMNPs compared to LMNPs was further validated by dynamic light scattering (DLS) measurements (Fig. 1c and Supplementary Fig. 1b). The elemental mapping confirmed the co-localization of elements, including gallium (Ga), indium (In), oxygen (O), phosphorus (P), and sulfur (S) throughout the RLMNPs (Fig. 1d), where P and S elements were derived from RAFT agents. The P signal of RAFT agents was also detected in Energy-dispersive X-ray spectroscopy of RLMNPs (Supplementary Fig. 2). In comparison, as expected, no P and S signals were found in the LMNPs (Supplementary Fig. 1c and d). The zeta potential of RLMNPs dispersed in water was changed from $29.94 \pm 3.21$ mV to $-0.01 \pm 3.89$ mV after the coating of RAFT agents (Fig. 1e). Furthermore, absorption peaks at ~310 nm in both RAFT agents (309 nm) and RLMNPs (311 nm) were observed in UV-Vis spectra, which confirm the presence of the thiocarbonyl groups of RAFT agents (Fig. 1f)[38]. C–H stretch and C=O stretch signals on both RAFT agents and RLMNPs were observed using Fourier-transform infrared spectroscopy (FTIR) (Fig. 1g). X-ray photoelectron spectroscopy analyses demonstrate the presence of P signals originating from RLMNPs, as indicated in Supplementary Fig. 3. All these data demonstrated that RAFT agents with diphosphonic acid groups successfully grafted onto the surface of LMNPs to generate the RLMNPs for photoinitiated 3D printing. Most importantly, RLMNPs have the potential to be prepared directly within 3D printing inks (such as monomers used for RAFT polymerization) rather than ethanol, enabling their homogeneous integration into the printing process.

### Fabrication and characterization of 3D-printed LMPCs

RLMNPs were utilized to fabricate LMPCs via using a stereolithographic 3D printer ($\lambda_{max} = 405$ nm, 0.81 mW/cm²) for subsequent performance in 4D printing. Specifically, LMPCs were printed through a type I photoinitiated RAFT polymerization by using tert-butyl acrylate (TBAm) as the primary monomer, poly(ethylene glycol) diacrylate (PEGDA, $M_n = 250$ g/mol) as the crosslinker, RLMNPs as the RAFT agent with a molar ratio of [RAFT agent]:[Diphenyl (2,4,6-trimethyl benzoyl) phosphine oxide (TPO)]:[TBAm + PEGDA] = 1: 0.25: 200 and a fixed percentage of the weight (wt%) of [TBAm]: [PEGDA] = 65: 35 (Fig. 2a). The Norrish Type I photoinitiator, TPO, was activated under violet light irradiation ($\lambda_{max} = 405$ nm) to facilitate polymerization in the 3D printer. The type I photoinitiator-RAFT mediated 3D printing method is characterized by its fast-building speed, high adaptability, and ability to produce high-resolution objects in a robust 3D printing environment[39]. As a result, LMPCs with simple shapes and morphologies, including cylinder, square, cuboid, and standard specimens for tensile test, were successfully printed (Fig. 2a). The slight shrinkage in the diameter of 3D-printed objects compared to their digital model (Supplementary Fig. 4) is a typical result of the radical photopolymerization process[39,40]. Scanning electron microscopy (SEM) and elemental mapping observed a smooth surface without any defects and the presence of essential elements (Ga, In, P and S) in the 3D-printed materials (Fig. 2b). In contrast, the S and P elements could not be detected in 3D-printed LMNP-polymer composites without RAFT

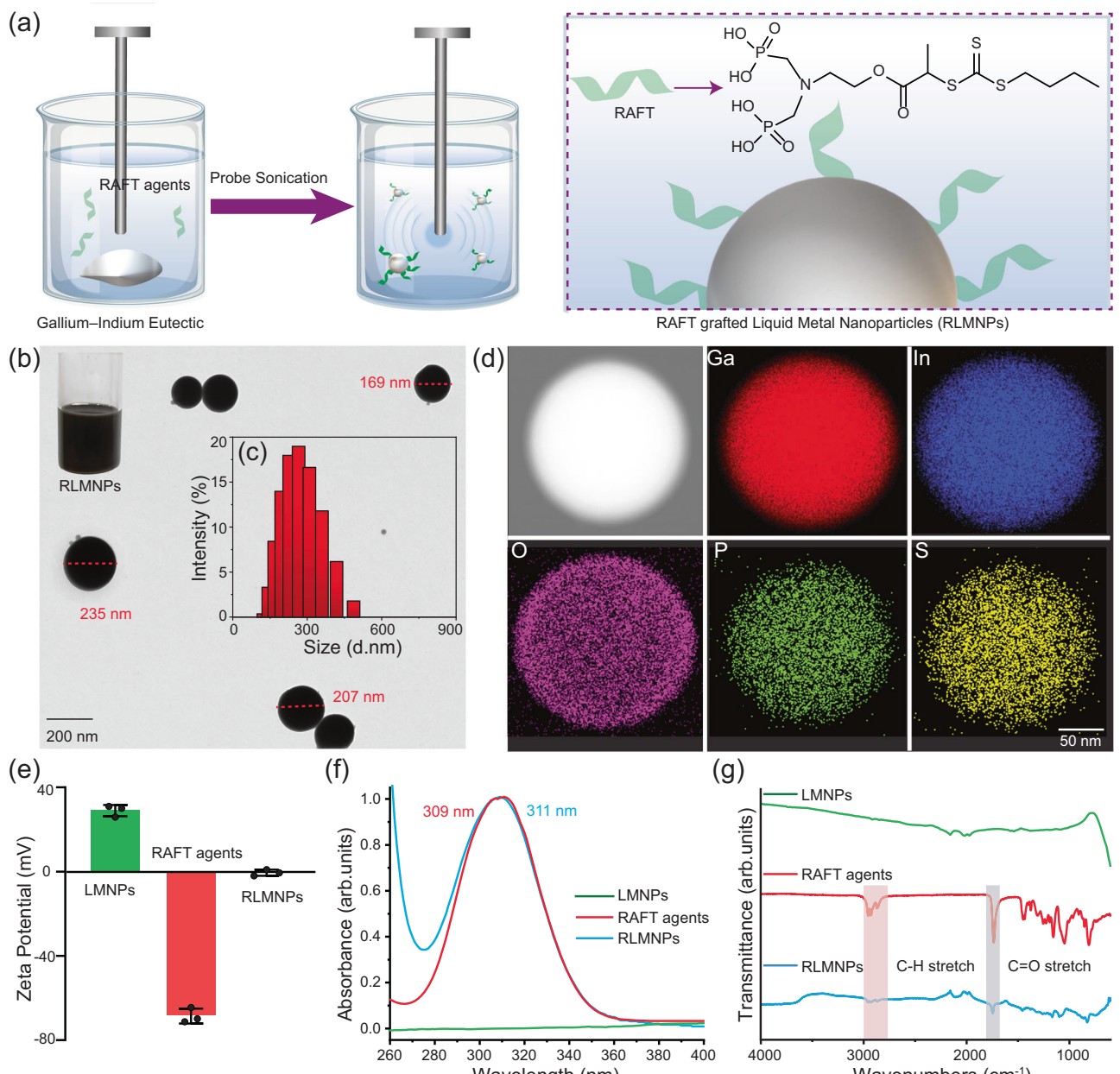

**Fig. 1 | The preparation and characterization of reversible addition–fragmentation chain-transfer (RAFT) agents grafted liquid metal nanoparticles (RLMNPs). a** The fabrication of RLMNPs in an ethanol solution via ultrasonication; (**b**) Photograph and the TEM image of RLMNPs; (**c**) Intensity-based size distribution histograms of RLMNPs measured by DLS; (**d**) Elemental mapping of RLMNPs, including Ga, In, O, P and S elements; (**e**) Zeta potential of liquid metal nanoparticles (LMNPs), RAFT agents and RLMNPs; Data were expressed as means ± SD. Error bars indicated standard deviations for 3 independent LMNP nanoparticle solutions; (**f**) UV-vis spectra, and (**g**) FTIR spectra of LMNPs, RAFT agents and RLMNPs.

agents due to the lack of RAFT agents (Supplementary Fig. 5). SEM imaging in Fig. 2c unequivocally depicted RLMNPs, affirming their successful integration within the 3D-printed polymers.

One of the advantages of integrating RAFT agents into 3D printing is to improve the resolution and accuracy of fabricated materials[39]. Under a consistent layer cure time of 40 s, objects with satisfactory resolution were successfully fabricated (Supplementary Fig. 6a). Conversely, regions adjacent to RAFT-free 3D-printed objects exhibited premature curing in identical conditions, resulting in the formation of polymerized masses and consequent degradation of object resolution (Supplementary Fig. 6b). Microscopic characterization was employed to further evaluate the resolution of 3D-printed objects. The SEM image showcased clear pore structures of the 3D-printed porous

object with RAFT agents (Supplementary Fig. 6c). Notably, these hollows were absent in the RAFT-free 3D-printed object (Supplementary Fig. 6d). RAFT agents would be regarded as a light-absorbing dye, effectively mitigating light scattering and thereby enhancing print resolution across all spatial axes[39]. Additionally, we employed RLMNPs to fabricate intricate designs resembling the iconic Sydney Opera House (Fig. 2d) and snowflakes (Supplementary Fig. 7) with exceptional structural fidelity. This highlights the benefits of incorporating RAFT agents into 3D printing, particularly in addressing the exacting resolution requirements of sophisticated structures in 4D-printed soft robots.

Additionally, we analyzed the mechanical properties of the as-printed LMPCs by comparing them to a control sample, which was a

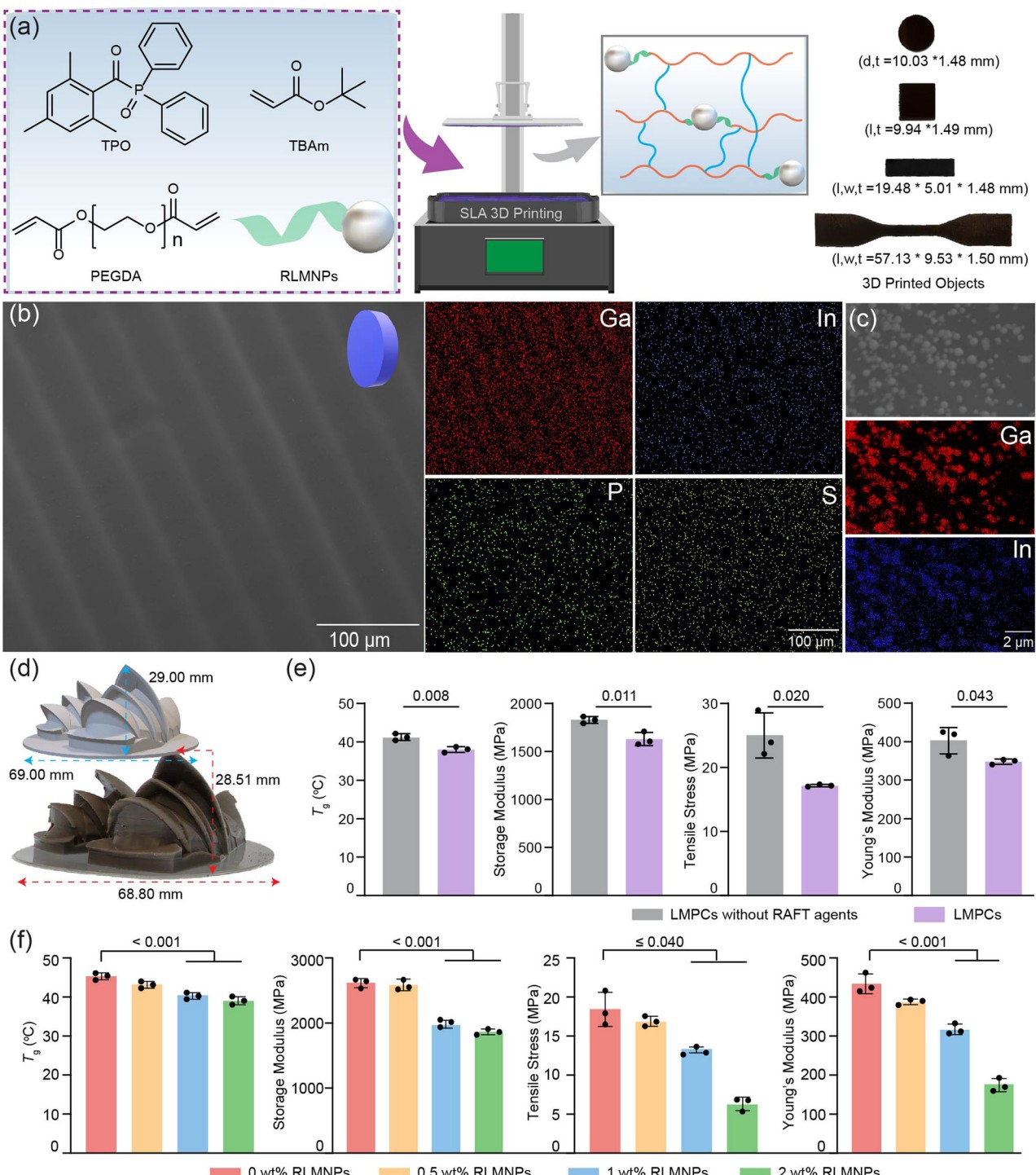

**Fig. 2 | 3D printing and characterizations of liquid metal polymer composites (LMPCs). a** The fabrication process of 3D-printed objects by using diphenyl (2,4,6-trimethyl benzoyl) phosphine oxide (TPO) as photo-initiator, *tert*-butyl acrylate (TBAm) as the monomer, poly(ethylene glycol) diacrylate (PEGDA) as a crosslinker, and RLMNPs as RAFT agents. d, t, l, and w mean the diameter, thickness, length and width of the LMPCs, respectively; (**b**) SEM photograph and EDS elemental mapping of 3D-printed objects with RLMNPs; the detected elements included Ga, In, P and S; (**c**) SEM photograph and EDS elemental mapping of RLMNPs in 3D-printed LMPCs;

(**d**) A model resembling the Sydney Opera House (1 wt% of RLMNPs) was manufactured using stereolithography 3D printing; (**e**) Mechanical properties of LMPCs and LMPCs without RAFT agents; (**f**) Mechanical properties of LMPCs containing different concentrations of RLMNPs (0, 0.5, 1, and 2 wt%). Bars in (**e** and **f**) represent means ± SD ($n = 3$ independent 3D-printed samples). Exact $p$-values given in (**e**) were obtained using an unpaired two-sided $t$ test, and $p$-value ranges provided in (**f**) were obtained using one-way ANOVA.

3D-printed LMPCs but without a surface coating of RAFT agents. Mechanical parameters of 3D-printed objects, including G′, $T_g$, σ and $E$, were analyzed using dynamic mechanical analysis (DMA) and a tensile testing machine. The G′ was taken at the temperature closest to 20 °C.

As shown in Fig. 2e and Supplementary Fig. 8, G′ and $T_g$ values of 3D-printed LMPCs were similar to those of the control sample (without RAFT agent), indicating comparable thermal parameters. However, σ, and $E$ were noticeably reduced from 24.66 ± 2.86 MPa and

$412.41 \pm 28.96$ MPa to $17.13 \pm 0.15$ MPa and $351.35 \pm 5.59$ MPa, respectively, indicating that the LMPCs are relatively softer than the control sample. Therefore, surface grafting with RAFT agent has a significant impact on network homogeneity and weak interlayer bonding[41], leading to reduced tensile properties of 3D-printed objects.

To understand the impact of RLMNPs content on the mechanical properties of LMPCs, we printed rectangular prisms ($20 \times 5 \times 1.5$ mm) with different weight % (0, 0.5, 1, and 2 wt%) RLMNPs for mechanical property measurements. Following the completion of 3D printing, all samples were placed to violet light irradiation ($\lambda_{max} = 405$ nm, 3.6 mW/cm$^2$) for 15 min to confirm full monomer conversions. The cure time per bottom layer and printed layer with the thickness of 50 μm in the process of 3D printing were prolonged from 25 s to 30 s and 22 s to 30 s, respectively, while the usage of RLMNPs increased to 2 wt% from 0.5 or 1 wt% (Supplementary Table 1). The RLMNPs with 2 wt% in resins can hinder the penetration of UV light, leading to increased cure time for each layer[42]. We also observed some breakages around the 3D-printed objects with 2 wt% of RLMNPs. Previous studies have shown that the incorporation of rigid nanoparticles into polymeric materials can enhance the $T_g$ and mechanical strength[42-45]. Because the incorporation of nanoparticles creates a physical barrier that limits molecular motions of polymers, leading to an increase in the $T_g$ of composites[46]. However, our study revealed that the incorporation of 1 and 2 wt% RLMNPs into a 3D-printed polymer matrix resulted in a statistically significant reduction in the mechanical strength (Fig. 2f). This was evidenced by a reduction in parameters such as G', $T_g$, σ, and E, as determined through DMA and tensile testing of standard test pieces. In comparison to the other groups, relatively lower $T_g$ values ($40.54 \pm 0.71$ °C and $39.31 \pm 0.87$ °C, respectively) were achieved with 1 and 2 wt% loading of RLMNPs (Supplementary Fig. 9). The soft and pliable nature of RLMNPs provides less resistance to the motion of polymers compared to rigid nanoparticles, thus resulting in a decrease in $T_g$. Tensile testing experiments also revealed a considerable drop in σ from $18.00 \pm 1.76$ to $6.37 \pm 0.72$ MPa and E from $434.38 \pm 20.63$ to $173.88 \pm 13.95$ MPa with 2 wt% of RLMNPs (Supplementary Table 1 and Fig. 10). We speculated that the reduction of the tensile is due to the soft and liquid characteristics of RLMNPs[47]. While the integration of 2 wt% RLMNPs into 3D-printed materials resulted in a relatively lower $T_g$, tensile stress, which may be favorable for activating shape-shifting in 4D printing, a concentration of 1 wt% RLMNPs was ultimately chosen for 4D printing after taking into account both printing efficiency and the integrity of the 3D-printed materials.

## One-step synthesis of RLMNPs-based resins for 3D printing

The conventional method for preparing nanoparticle-based resins for 3D printing involves two steps: the synthesis of nanoparticles, followed by nanoparticle immersion in resins for 3D printing. RLMNPs offer the additional advantage of being directly synthesized in solutions using a one-pot approach[48,49], thus minimizing EGaIn waste, saving time, and simplifying the fabrication process. In the current study, RLMNPs were directly prepared by sonicating bulk EGaIn in four liquid resins ([TPO]:[PEGDA]: [Monomer] = 0.125: 65: 35) containing different monomers N-hydroxyethyl acrylamide (HEAAm), 2-hydroxyethyl acrylate (HEAm), TBAm, and N, N-dimethyl acrylamide (DMAm) as the main component, respectively (Fig. 3a). According to the TEM images presented in Fig. 3b, all RLMNPs prepared in different resins displayed spherical shapes and similar diameters (about 100–200 nm). As shown in Fig. 3c, it was found that the average size of RLMNPs from DLS was slightly larger than those measured by TEM, due to the surface grafting of RAFT agents[50]. Subsequently, all of those RLMNP-based resins were successfully printed into standard tensile test specimens with comparable dimensions (Fig. 3a), indicating that our one-step printing is a universal approach for the fabrication of 3D-printed LMPCs. Compared to conventional 3D printing, one-step 3D printing offers several advantages such as simplification of the resin preparation process and

the ability to integrate multiple functionalities of nanomaterials into a single object, which can enable the development of materials for advanced applications. The higher photopolymerization efficiency of acrylamide monomer results in a slightly faster printing speed (4.2 μm/s) than that of TBAm and HEAm (3.2 μm/s). Faster printing speed in 3D printing is beneficial to increase productivity, reduce lead times, and mitigate aggregation and precipitation of nanomaterials during printing, resulting in improved product stability and uniformity. DMA and tensile tests revealed the selection of monomers in resins had a significant impact on the mechanical properties of printed objects (Fig. 3d, Supplementary Table 2, Fig. 11 and 12). For example, HEAAm-based objects demonstrated superior mechanical properties, with the highest tensile stress ($46.77 \pm 3.11$ MPa) and storage modulus ($4190.48 \pm 24.27$ MPa) compared to materials generated by the other three monomers. This phenomenon can be attributed to amide and hydroxyl groups in HEAAm, which facilitate the formation of numerous intramolecular and intermolecular hydrogen bonds[51-53]. The substantial number of hydrogen bonds collectively and synergistically enhances the flexibility and tensile strength of 3D-printed materials[54]. Moreover, these hydrogen bonds continually undergo a process of breaking and reforming during tensile forces, further contributing to the material's mechanical properties[55]. Nanosized RLMNPs would be directly prepared in liquid resins with various ratios of monomer and crosslinker (Supplementary Table 3 and Fig. 13) to achieve composites with different mechanical strengths (Supplementary Fig. 14).

We further evaluated the cytotoxicity of the 3D-printed materials (Supplementary Fig. 15) on in vitro cell models using human embryonic kidney cells (HEK-293) and mouse macrophage cells (Raw 264.7). No significant toxicity was observed on both cell lines, indicating excellent biocompatibility of our 3D-printed materials. This underscores the potential applicability of the one-step LM nanoparticle-based resin printing process to other 3D printing techniques, including masked stereolithography, bioprinting, organ and tissue printing.

## Photothermally-responsive 4D printing of LMPCs

In addition to their exceptional fluidity, EGaIn-based nanoparticles possess superior photothermal transduction properties[56,57], which can serve as a stimulus for creating soft materials with thermal-responsive characteristics. Therefore, we further investigated the photothermal property of the 3D-printed LMPCs by using an infrared camera for real-time monitoring of the temperature of printed materials under NIR light irradiation. 3D-printed items with different concentrations of RLMNPs (0, 0.5, 1, and 2 wt%) were exposed to NIR laser irradiation ($\lambda_{max} = 808$ nm, 0.3 W/cm$^2$) for 180 s. As shown in Supplementary Fig. 16a, the temperature of 3D-printed materials with 1 wt% RLMNPs dramatically increased from 25 °C to 78 °C, whereas no obvious temperature change was observed in 3D-printed polymeric materials without RLMNPs. Thus, it can be inferred that the integration of LMNPs into 3D-printed objects can significantly enhance their photothermal transduction efficiency, which may lead to effective NIR light-responsive 4D printing. Furthermore, as depicted in Supplementary Fig. 16b, raising the laser intensity results in a temperature rise of the 3D-printed materials (1 wt% of RLMNPs) due to the additional energy. However, to assess the responsiveness of LMPCs to NIR light for 4D printing purposes, a moderate laser intensity of 0.3 W/cm$^2$ was chosen.

The exceptional photothermal properties of 3D-printed LMPCs with 1 wt% of RLMNPs inspired us to develop innovative NIR light-mediated 4D printing technology. Typically, LMPCs ($l \times w \times h = 59.13 \times 9.53 \times 1.51$ mm) were bent into a predetermined shape at a bending angle for 70° in hot water and cooled down to main the programmed shape. Upon triggering under the NIR stimulus, the shape recovery performance of the curved specimen was evaluated (Fig. 4a). As shown in Fig. 4b, when subjected to laser irradiation ($\lambda_{max} = 808$ nm, 0.3 W/cm$^2$), the temperature of LMPCs exceeded $T_g$ by approximately 40 °C within just 180 s. This observation indicates that the composites hold

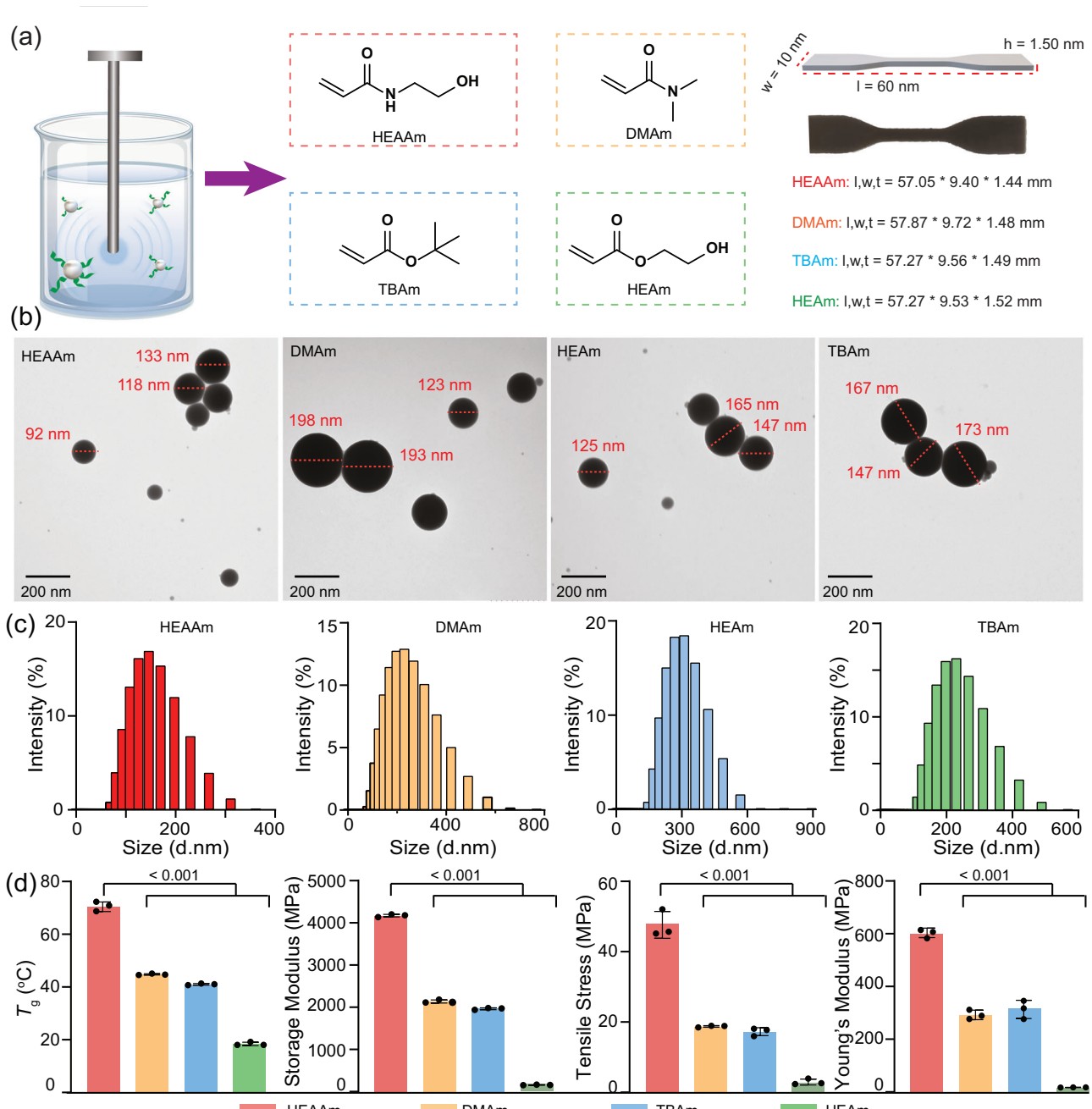

**Fig. 3 | Fabrication and characterization of LMPCs using different monomers.** **a** RLMNPs were prepared in different liquid resins using N-hydroxyethyl acrylamide (HEAAm), 2-hydroxyethyl acrylate (HEAm), *tert*-butyl acrylate (TBAm), and N, N-dimethyl acrylamide (DMAm) as main monomers, respectively. l, w, and t mean the length, width, and thickness of the LMPCs, respectively; **b** TEM photographs of RLMNPs in liquid resins with different monomers. The experiments were repeated independently three times with similar results; **c** Intensity-based size distribution histograms of RLMNPs in liquid resins with different monomers measured by DLS. The size of RLMNPs was measured immediately after sonication; (**d**) Mechanical properties of 3D-printed LMPCs using different monomers and the storage modulus of 3D-printed objects was determined at 20 °C. Bars in (**d**) represent means ± SD ($n = 3$ independent 3D-printed samples). The $p$-value ranges provided in the graphs were obtained using one-way ANOVA.

promise for NIR-mediated shape recovery. Movies were captured to monitor the recovery angles of 3D-printed objects in real time during laser irradiation (Supplementary Movie 1). As shown in Fig. 4c, three stages in the recovery process were recorded when LMPCs with a predetermined shape were exposed to NIR laser irradiation at 0.3 W/cm². In the initial stages (0–5 s), the shape recovery process was either absent or was sluggish due to the high friction forces between molecules, which hindered the release of the stored force[58]. As the temperature of the specimen approaches $T_g$, the material gradually starts to recover its original shape by releasing the stored force. In the final

stages (50–60 s), the shape recovery rate slowed down as the majority of stored force had already been released. It is important to note that in the control groups, where specimens did not have RLMNPs or were not exposed to NIR light, no shape recovery was observed. Fig. 4d directly depicted the recovery process of LMPCs from a bending shape of 70° to an almost straight morphology of 179° after 55 s of laser irradiation, demonstrating an almost 99% shape recovery ratio. It was observed that the time taken to return to the original shape was further reduced with the use of a 0.5 w/cm² laser intensity compared to 0.3 w/cm² (Supplementary Fig. 17). Interestingly, the shape recovery process

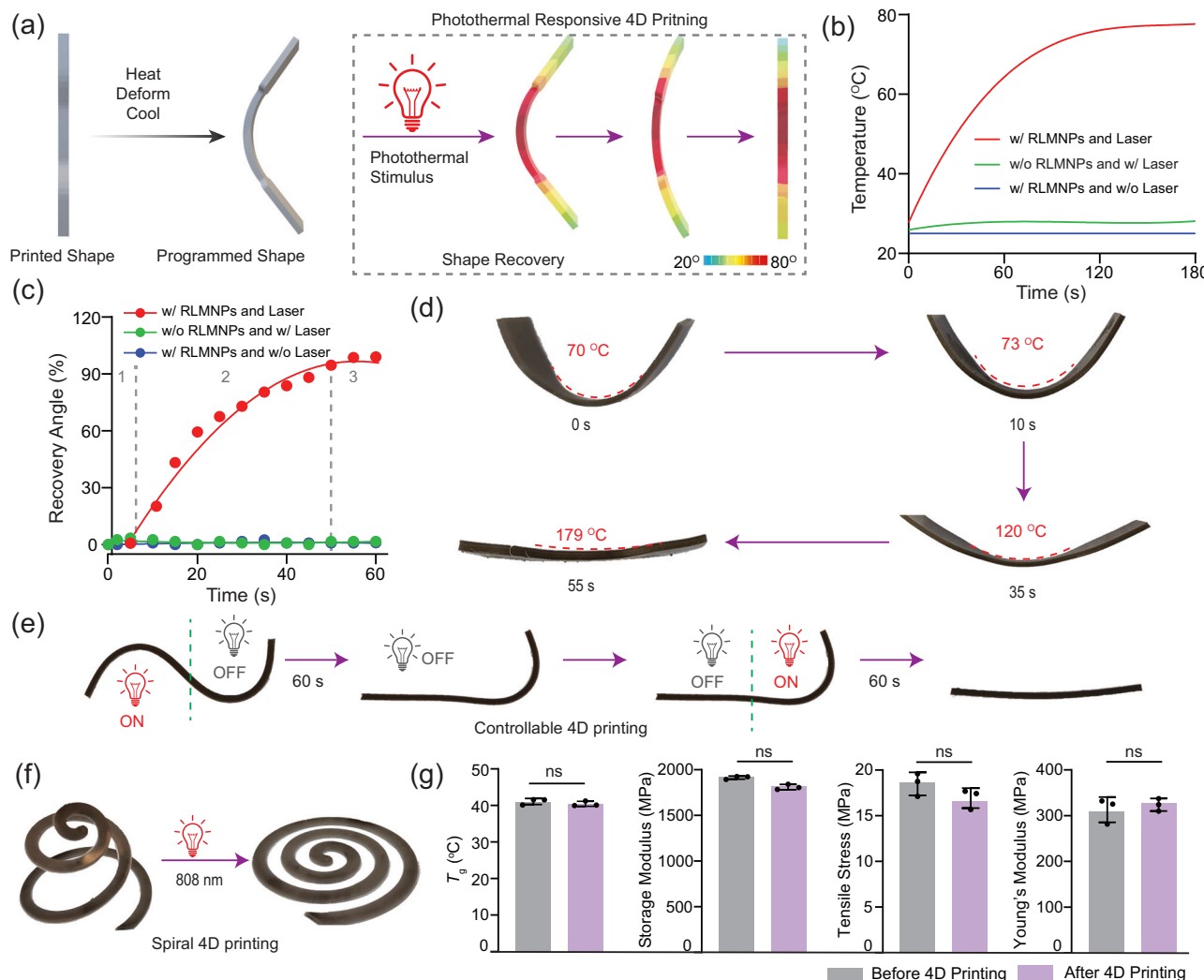

**Fig. 4 | NIR-responsive 4D printing of LMPCs with 1 wt% RLMNPs. a** The diagram of photothermally induced 4D printing; (**b**) Photothermal effect and (**c**) the recovery angle (%) on the time of 3D-printed composites; The line graph in red denotes 3D-printed LMPCs under the laser irradiation ($\lambda_{max}$ = 808 nm, 0.3 W/cm$^2$); The line graph in green denotes 3D-printed objects without RLMNPs under the same laser irradiation; The line graph in blue denotes 3D-printed LMPCs in darkness; (**d**) The shape recovery process of 3D-printed LMPCs under laser irradiation ($\lambda_{max}$ = 808 nm, 0.3 W/cm$^2$) for 60 s; (**e**) The controllable shape recovery of 3D-printed materials by switching ON/OFF the laser; (**f**) The shape recovery process of spiral objects under laser irradiation ($\lambda_{max}$ = 808 nm, 0.3 W/cm$^2$); (**g**) Mechanical properties of LMPCs before and after performing NIR-responsive 4D printing. Bars in (**g**) represent means ± SE ($n$ = 3 independent 3D-printed samples). Statistical significance is calculated based on an unpaired two-sided $t$-test and ns means not significant.

could be controlled by switching ON/OFF the laser light (Fig. 4e). In addition to basic models, the programmed shape of 3D objects could be recovered to their 2D original pattern after being irradiated with a laser ($\lambda_{max}$ = 808 nm, 0.3 W/cm$^2$, Fig. 4f, and Supplementary Fig. 18). No significant changes were observed in the mechanical properties of LMPCs (e.g., $T_g$, 41.18 ± 0.68 °C $vs$ 40.53 ± 0.56 °C; Tensile stress, 18.82 ± 1.27 MPa $vs$ 16.90 ± 1.09 MPa) during the shape recovery process (Fig. 4g and Supplementary Fig. 19), indicating remarkable structural stability during the 4D printing.

Subsequently, we conducted a comparative analysis to evaluate the 4D performance of composites prepared by using rigid and liquid nanoparticles, respectively. In this context, iron oxide nanoparticles (IONPs) were selected for their similar photothermal conversion efficiency to LMNPs[59]. We observed that both types of composites exhibited a comparable temperature increase under laser irradiation for 180 s ($\lambda_{max}$ = 808 nm, 0.3 W/cm$^2$, Supplementary Fig. 20a). However, the LMPCs outperformed the IONP-based composites in terms of shape recovery capacity, achieving a recovery angle of 99% compared to 91% in IONP-based composites (Supplementary Fig. 20b), which can

be attributed to the softness of LMNPs in contrast to the rigidity of IONPs.

### Evaluation of repeatability and weightlifting of LMPCs
To explore the potential applications of RLMNPs-based 4D printing materials, we conducted recycling and weightlifting experiments on LMPCs. Repeatability, as an important aspect of shape memory materials, was evaluated by performing the 4D printing of LMPCs under repeated laser irradiation, as illustrated in Fig. 5a. Photothermal results showed that the maximum temperature rose during the first 60 s and photothermal curves of composites remained consistent even after repeated laser irradiation (Supplementary Fig. 21), supporting the testing of the repeatability of NIR-mediated 4D printing. Importantly, the recovery angles of LMPCs remained unaffected through a minimum of 25 cycles of programming and NIR light irradiation as depicted in Fig. 5b, demonstrating robust shape memory recyclability. SEM photographs confirmed the stable nanostructures of RLMNPs in 3D-printed materials (Supplementary Fig. 22) and mechanical tests indicated similar tensile strength over 25 cycles

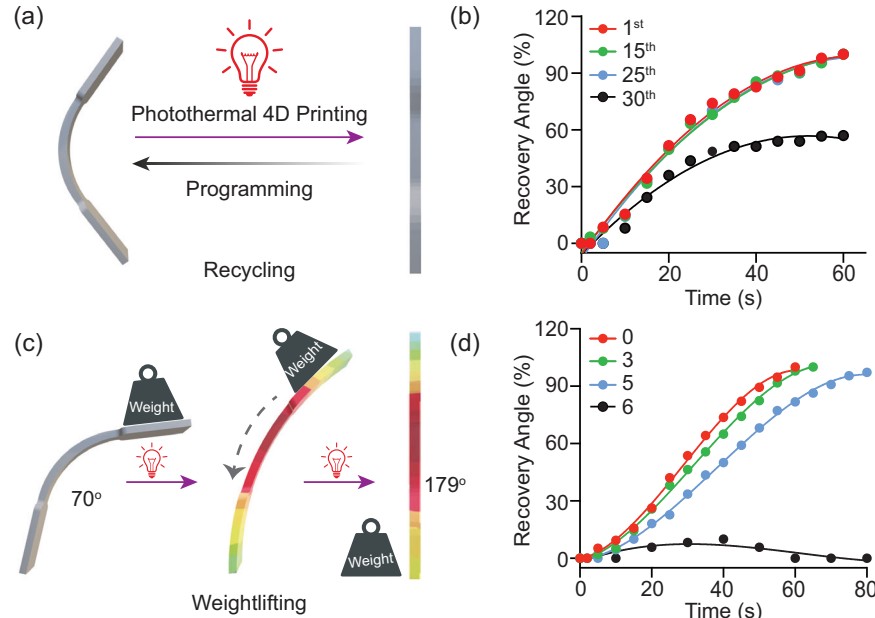

**Fig. 5 | Potential applications of NIR-responsive 4D printing with 1 wt% RLMNPs. a** The recycling process of shape memory polymers in repeated programming and photoinduced 4D printing; (**b**) The recovery angle on time of LMPCs in a total of 25 cycles while irradiating with 808 nm laser (0.3 W/cm²) for 60 s; (**c**) The weightlifting process induced by 4D printing; (**d**) The recovery angle on time of 3D-printed LMPCs under 808 nm laser irradiation when placing items with different multiples their weight (0, 3, 5 and 6) on the surface.

(Supplementary Fig. 23). When repeated 30 times, recovery efficiency notably dropped to about 57%, alongside the reduced (25 %) σ and (20%) $E$ of the original object (Supplementary Fig. 24). The excellent recycling capability of RLMNPs-based 4D printing presents significant potential for enabling energy and cost-efficient robots, aircraft, and ships.

Moreover, the weightlifting capability of 3D-printed materials presents another valuable application. Typically, when weight is placed on the printed material in the direction of recovery, the shape recovery process causes 3D-printed materials to lift their weight, as shown in Fig. 5c. LMPCs were able to rapidly lift 5 times their weight under laser irradiation ($\lambda_{max}$ = 808 nm, 0.3 W/cm²) while maintaining similar shape recovery performance, as depicted in Fig. 5d. The exceptional photo-induced weightlifting ability allows 3D-printed LMPCs to open up opportunities for their application in soft robotics.

### Application of the 3D-printed LMPCs as NIR-responsive soft robots

Harnessing the rapid photothermal responsiveness of LMPCs accompanied by a large deformation capacity and robust force generation, we explored the use of LMPCs for the development of soft robotic systems. As shown in Fig. 6a, we designed and printed a flower petal-like soft robot comprised of LMPCs, which were subsequently programmed to a temporary shape capable of grasping a cap. Upon NIR irradiation ($\lambda_{max}$ = 808 nm, 0.3 w/cm²) at the top of the robot, it rapidly returned to its pre-deformation state, analogous to the unfurling of flower petals, resulting in the release of the cap (Fig. 6b). Nevertheless, in the absence of irradiation, the programmed soft robot consistently maintained its grip on the object (Supplementary Movie 2).

In another scenario, object grasping, and release can be achieved by directing irradiation to specific positions on the 3D-printed soft robots (Fig. 6c). Following a 40 s exposure to NIR light at the base region of the programmed device, a rapid transition from a curved to a straightened state occurred. This deformation caused the bottom grippers to close, successfully capturing the cap. Subsequently, by adjusting the NIR light to irradiate the upper region of the device, a gradual unfolding of the entire material was observed, resulting in the release of the held object (Fig. 6d and Supplementary Movie 3).

Additionally, LMPCs' shape memory capability, coupled with the 3D printing technology developed in this study, empowers the creation of bistable structures tailored for locomotion in soft robotics. In this case, rotational bistable components were connected by two 3D-printed tuning arms comprised of LMPCs (l × w × h = 10 × 2 × 1 mm). As shown in Fig. 6e, the inner part can be inter-rotated into two distinct stable states (A and B), while the outer ring remains fixed in place. At room temperature, the tunning arm is curved due to the programmed LMPCs, and the inner part is rotated 21° to the left, maintaining stable State B. Upon 30 s of NIR irritation, the deformed tuning arms returned to their pre-deformation state, and the inner part shifted 21° to the right, transforming into stable State A (Fig. 6f). Consequently, by employing LMPCs elements, we demonstrate the feasibility of tuneable bistable structures as soft actuators for mechanical switches, showcasing promising applications in the medical device sector, consumer electronics, and the automotive industry.

In summary, we have successfully developed a universal synthetic approach that integrates RLMNPs/LMNPs into various 3D-printed resins to fabricate LM-polymer composites with high-resolution and complex structures. The 3D-printed LMPCs are demonstrated with decreased mechanical strengths when compared to 3D-printed polymers due to their unique soft and liquid characteristics of RLMNPs. On the other hand, the excellent photothermal properties of RLMNPs enabled the shape memory behavior for NIR light-mediated 4D printing. The 3D-printed LMPCs showed remarkable repeatability for up to 25 shape memory processes, with no significant reduction in shape recovery ratio, and exhibited remarkable weightlifting capacity during shape recovery. More importantly, the 3D-printed LMPCs have great feasibility for soft robots, which are capable of grasping and releasing objects under NIR stimuli and enable precise manipulation of rotational bistable structures. These findings demonstrate the promising potential of LMPCs-based 4D printing in soft robots, light-driven materials, medical tools, and actuators, emphasizing the importance of further research in these fields.

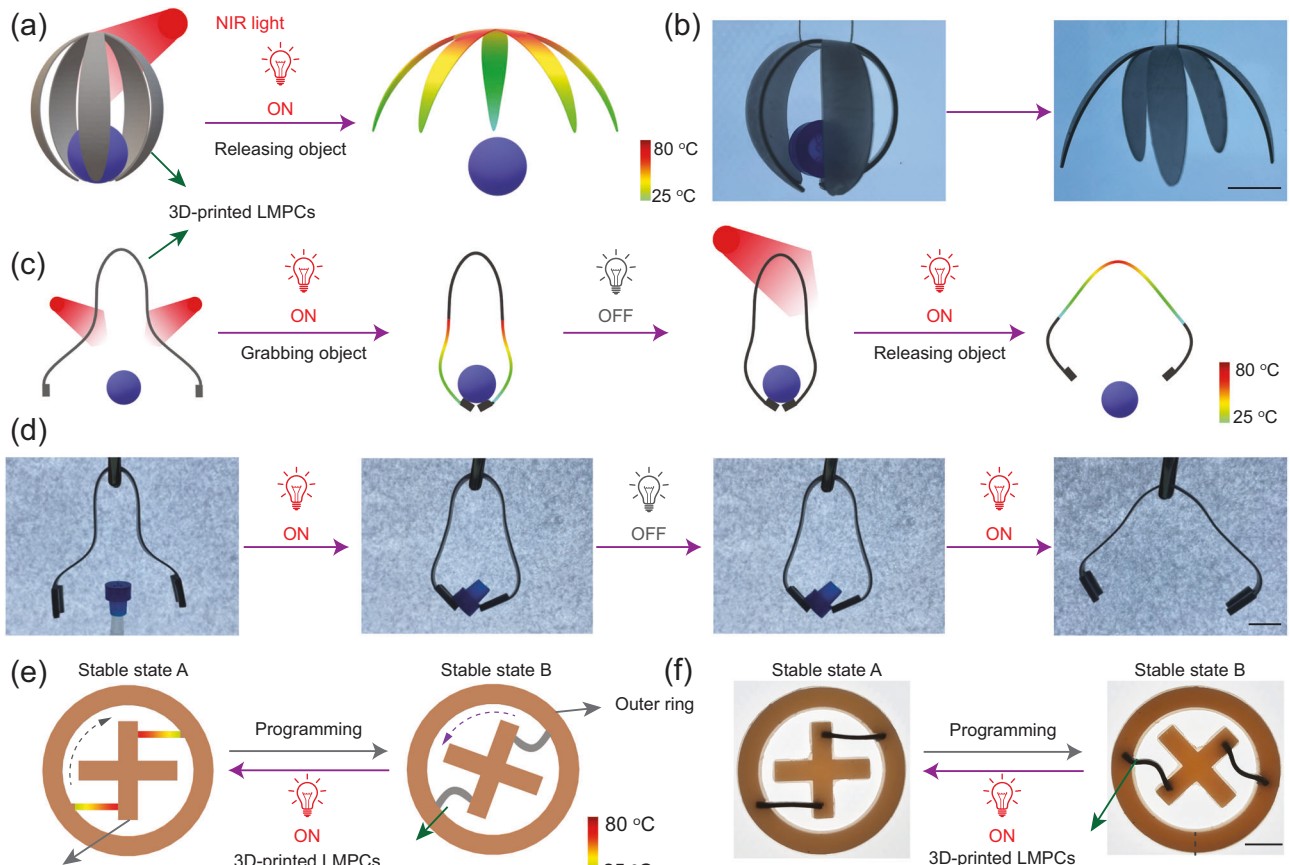

**Fig. 6 | Application of the 3D-printed LMPCs as near-infrared (NIR)-responsive soft robots. a** The diagram and (**b**) demonstration of releasing an object via NIR light responsive LMPCs. The scale bar corresponds to 10 mm (**c**) The diagram and (**d**) demonstration of gripping and releasing an object via using NIR light controlled LMPCs. The scale bar corresponds to 10 mm (**e**) The diagram and (**f**) demonstration of the rotated structure (state B) transforms into the original shape (state A) upon NIR exposure. The scale bar corresponds to 5 mm.

## Methods

### Materials
Gallium–Indium eutectic (EGaIn, Ga 75.5% and In 24.5%, ≥99.99%), *tert*-butyl acrylate (TBAm, 98%), 2-hydroxyethyl acrylate (HEAm, 96%), N-hydroxyethyl acrylamide (HEAAm, 98%), N, N-dimethyl acrylamide (DMAm, 99%), poly(ethylene glycol) diacrylate (PEGDA, $M_n$ = 250, PEGDA, >92%), and diphenyl (2,4,6-trimethyl benzoyl) phosphine oxide (TPO, >97%) were purchased from Sigma-Aldrich. All other reagents were used as received unless otherwise specified. Human embryonic kidney cells (HEK-293) and mice macrophage cells (Raw 264.7) were purchased from ATCC (USA).

### Instrumentation
The morphology and size of LM-based nanoparticles were observed by transmission electron microscopy (TEM, Hitachi HT7700 Bm and Hitachi HF5000 Cs-STEM/TEM system). The hydrodynamic sizes of LM nanoparticles were analyzed at 298.0 K using Zetasizer Ultra (Malvern) with ZS XPLORER software (v2.0.1.1). All measurements were repeated three times. UV-vis spectra were recorded using a CARY 300 spectrophotometer (Agilent) with UVProbe software (v2.42). FT-IR spectra were recorded on a Nicolet 6700 spectrometer under attenuated total reflectance (ATR) with OMNIC software (v8.3.103). The 3D printing was performed by using a photo 3D printer (Photon Mono SE, Anycubic). The targeted material geometries and .stl files were generated using Cinema 4D-Maxon, and printing parameters were generated using PhotonWorkshop (2.2.15). Scanning electron microscopy (SEM) of 3D-printed objects was performed on a JEOL JSM-7100F and elemental

mapping of 3D-printed objects was analyzed by using SEM microscopy (HITACHI SU3500 SEM-EDS). Thermal and mechanical properties were analyzed using a TA Instruments Mettler Toledo dynamic mechanical analyzer, with STARe Software (v9.10) for data collection. A single-column tensile testing machine (YK-Y0084, Dongguan Yaoke Instrument Equipment Co., Ltd., China) was used to measure the tensile stress (σ) and Young's modulus (E) of samples at a rate of 5 mm/min. Data was collected by using TM2101 software (v5.58). The toxicity of 3D-printed objects was measured using the multimode plate reader (EnSight, PerkinElmer, Inc.) with Kaleido software (v3.0.3067.117). Photothermal experiments and the rest of the recovery angle were carried out by an 808 nm stabilized infrared fiber laser system (Leoptics, Shenzhen LEO-Photoelectric Co., Ltd).

### Preparation of LMNPs and RLMNPs
75 mg of EGaIn was added in 15 mL of ethanol and put in the ice bath for 30 min to cool down. Afterward, the EGaIn solution was sonicated under an ice bath for 25 min (30% power, Sonics VCX-750 Vibra Cell Ultra Sonic Processor equipped with a 6 mm sonication probe). After sonication, the mixture was regarded as the LMNPs solution.

50 mg of RAFT and 75 mg of EGaIn were mixed in 15 mL of ethanol and put in the ice bath for 30 min to cool down. Afterward, the EGaIn solution was sonicated under an ice bath for 25 min. The supernatant was collected to measure the efficiency of RAFT agent attachment to the surface of LMNPs. The precipitation was collected and dispersed into the solution by water bath sonication to achieve RLMNPs solution.

## Fabrication of 3D-printed LMPCs

A typical procedure for 3D printing is as follows: 10.5 mL of TBAm (73.2 mmol, 9.3 g,), 4.5 mL of PEGDA (20 mmol, 5.0 g), 46 mg of TPO (0.12 mmol), and previously prepared RLMNPs that containing 75 mg of LMNPs and 246.8 mg of RAFT (0.47 mmol) were added into a 20 mL glass vial to obtain a mixed solution with a molar ratio of [RAFT]: [TPO]: [TBAm+PEGDA] = 1: 0.25: 200 and a mass ratio of [TBAm]: [PEGDA] = 65:35. The reaction mixture was sonicated for 20 min under ice bath, prior to addition to the 3D printer vat, and subsequently irradiated with spatially controlled violet light ($\lambda_{max}$ = 405 nm, 0.81 mW/cm$^2$) during the 3D printing process. The targeted objects (disk, diameter × thickness = 15.00 × 1.50 mm; cube, length × height = 10.00 × 2.00 mm; cuboid, length × width × height = 20.00 × 10.00 × 1.5 mm; and Samples for tensile testing, length × width × height = 60.00 × 10.00 × 1.5 mm) and .stl files were produced using Cinema 4D - Maxon, and printing parameters including layer cure times and slicing thickness were generated using the photon workshop software. The first 5 (bottom) layers of the disk were irradiated for 25 s to make sure adhesion between the 3D-printed material. The regular cure time (23 s) per layer as stated in the main text applies. The Z lift distance was 1 mm, the Z lift speed was 3 mm/s, and the Z retract speed was 3 mm/s. After the object was printed, the surface of the printed objects on the build stage was lightly washed with ethanol to remove the mixture solution. 3D-printed LMPCs using different monomers were fabricated by utilizing consistent methodologies.

## The photothermal effect of LMPCs

RLMNPs-based rectangular specimens were printed to test photothermal effects. Samples were irradiated with 808 nm NIR laser at different laser power densities (0, 0.1, 0.3, and 0.5 w/cm$^2$) for 3 min. The maximum temperature of the specimen was monitored in real-time by the infrared thermal imaging system. Additionally, the recycling test of the photothermal effect was performed by switching ON and OFF the laser irradiation. 3D-printed samples were irradiated with 808 nm NIR laser (0.3 w/cm$^2$) for 2 min and then placed in the dark for 2 min. This process is repeated five times to observe the photothermal effect. The maximum temperature of the specimen was monitored in real-time by the infrared thermal imaging system.

## Photothermally-responsive 4D printing of LMPCs

An LMPCs with l × w × h = 59.13 × 9.53 × 1.51 mm was printed by a 3D printer. The specimen was immersed into a hot water bath with a certain temperature exceeding $T_g$ about 20 °C for 5 min and exerted an external force to get a sample with a bending angle (70 °C). After that, the deformed specimen was fixed by quickly immersing it in cold water, and then the external force was removed. For photoinduced 4D printing, 3D-printed samples with a bending angle (70°) were irradiated with 808 nm NIR laser (0.3 W/cm$^2$) for 60 s and a camera was used to record the movie of the recovery process. The shape recovery angle and time were achieved by analyzing the recorded movie of the recovery process. A range of related 4D experiments, encompassing cycling tests, lifting, releasing, and grasping items, as well as controlling rotational bistable structures, were conducted under similar experimental conditions.

## Biotoxicity test of 3D-printed materials

HEK-293 and Raw 264.7 were seeded in 24 well plates and co-cultured with 3D-printed materials for 24 h. Afterward, 50 μL AlamarBlue reagent was added to each well. Absorbance was measured at 570/600 nm after 3 h incubation using the EnSight plate reader.

## Statistics and reproducibility

The data in all histograms are expressed as means ± standard deviation (SD) with statistical significance assessed by unpaired two-sided $t$-test, and one-way ANOVA. GraphPad Prism (v10.0.2) was used for the data analysis.

## Data availability

The authors declare that the data supporting the findings of this study are available within the paper and its Supplementary Information files. Additional data are available from the corresponding author upon request. Source data are provided as a Source Date file. Source data are provided with this paper.

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

## Acknowledgements

R.Q. would like to acknowledge the support from the National Health and Medical Research Council (APP1196850), Queensland-Chinese Academy of Sciences (Q-CAS) Collaborative Science Fund (QCAS2022016), UQ Amplify Women's Academic Research Equity (UQAWARE), UQ Foundation of Excellence Research Award (UQFERA), and Advance Queensland Women's Research Assistance Program (AQWRAP). T.P.D. is grateful for the National Health and Medical Research Council (APP1197373 and 2019056). This work used the Queensland node of the NCRIS-enabled Australian National Fabrication Facility (ANFF), and the Centre for Microscopy and Microanalysis (CMM).

## Author contributions

L.Z., S.T., T. P. D. and R.Q. conceived the original concept and initiated this project. L.Z. designed the experiment, participated in the entire project, and wrote the article. X.H. assisted in the preparation and characterization of nanoparticles. H.L. assisted in designing the experiment on 4D printing. J.H., T.C. and S.T. characterized the mechanical properties of composites. W.L., S.T., C.B., T.P.D. and R.Q. provided critical feedback and helped revise this article.

## Competing interests

The authors declare no competing interests.
