## [Peer Review File · Nature Communications]

REVIEWER COMMENTS

Reviewer #1 (Remarks to the Author):

Recommendation: Reject

Main comment:

Zhang et al. grafted RAFT reagents onto liquid metal nanoparticles (RLMNPs) and successfully applied them to UV-mediated stereolithographic 3D printing and near-infrared light-responsive 4D printing. These findings highlight the potential of liquid metal-polymer composites in 4D printing and provide a promising path to developing advanced materials for soft robotics, wearables, and aerospace applications.

However, there are still some problems, such as the misunderstanding of the concept of 4D printing and the lack of highlights of this research.

Therefore, I think it is not enough to publish in Nature Communications.

- 1、 The author mentioned 3D/4D printing in the abstract. However, in the manuscript, the degree of aggregation of molecular chains in the upper and lower layers of materials caused by NIR illumination is different, resulting in deformation. True 4D printing has an extra dimension of time. What is the author's understanding of 4D printing ?
- 2、 So far, many researchers have used NIR to deform liquid metal polymers. What are your advantages compared to them ?
- 3、 Why can RAFT agents be grafted onto liquid metal surfaces? FITR and UV-vis characterization methods have limitations and are not sufficient to explain. Please provide more favorable evidence.
- 4、 The author said that the resolution of 3D printing has been improved, how much has the resolution been improved? Detailed proof is needed that the resolution has been increased.
- 5、 The resolution of SEM and mapping images is too low.
- 6、 The authors mention that one-step printing of liquid metal nanoparticle-based resins can be applied to other 3D applications, such as bioprinting. However, the authors did not explore its biocompatibility. TPO and PEGDA are both toxic substances, why can they be used in bioprinting?
- 7、 The authors mention that intramolecular hydrogen bonding through amide groups leads to increased flexibility and stretchability. Why are intramolecular hydrogen bonds produced instead of intermolecular hydrogen bonds? And confusingly, can hydrogen bonding enhance both tensile strength and flexibility?
- 8、 Deformation caused by NIR is not novel, and only 10 reciprocating recoveries are not sufficient to demonstrate stable recovery.
- 9、 What are the specific applications of RLMNPs-based materials? I think only the recovery of deformation and weightlifting is not enough to prove its application, this is only performance.

Reviewer #2 (Remarks to the Author):

This is an interesting and exciting new study showing a new method to create 4D printed structures that could be used as soft robots in the NIR region.

The study is very well done and has merit for publication. I have just a few minor remarks that would be needed to tackle before the manuscript is ready for publication:

1) It is not clear why the use of nanoparticles in other approaches would be a drawback in terms of altered mechanical properties or physical characteristics of the final composites. One could certainly modulate such properties to an extent, therefore being more of an advantage than a disadvantage. This is discussed in the introduction, but should be better elaborated.

2) I miss a final soft robotic application. In which application would the proposed liquid metal polymer composites be of use? The experiments performed beyond the more fundamental characterization of the new system aim at creating demonstrations more than real applications.

Reviewer #3 (Remarks to the Author):

The manuscript presents a new advanced material, a composite of soft and liquid nanoparticles and polymer, that can change shapes once by NIR irradiation. Instead of rigid nanoparticles that are conventionally used in such composites, soft and liquid particles are used. This gets rid of the undesired effects arising from the rigidity of the nanoparticles. The authors report that the soft inclusions reduce glass transition temperature, tensile stress, and modulus of the composite. These observations are opposite to what happens in case of rigid nanoparticles.

The manuscript is well written and the results can have significant implications in the field of soft robotics. However, it requires a major revision before publication in Nature Communications. In summary, a significant gap exists between the material manufacturing and its application in "soft robots" (a word that is in the title of the manuscript). The paper reads like a traditional materials science paper focusing on manufacturing and characterization without demonstrating it in soft robots. Detailed comments follow.

Quantitative and direct comparisons between the proposed composite (that uses liquid nanoparticles) and comparable composites with rigid particles should be provided. Otherwise, the reader is left wondering what's the point of using liquid and soft particles?

A major criticism is the potential application of the composite to build soft robots. Only two shapes (a bent bar and a coiled spring-like object) are shown. The shape change is rather slow. For publication in a top generalist journal like Nature Communications, I would expect the paper to include more practical demonstrations. Even if the authors cannot go all the way to building soft robots, they should be able to

show some simple demonstrations like grasping a light object (say, a pen) and releasing it once exposed to NIR.

Longer cyclic loading should be investigated. Fig. 4(g) includes data on material properties after one cycle of loading. But, what is the limit? When does the material stop functioning (i.e., loses its ability to recover original shapes even with NIR irradiation)?

The title could be improved or the manuscript could be augmented so that the title is indeed a summary of the paper. The title gives the sense that a “3D-printed liquid metal polymer composite,” which is a type of material, is equivalent to “NIR-response 4D printing soft robot, which is a manufacturing technique. As mentioned earlier, not much is discussed about the application of the material in soft robots.

Line 20: The abbreviation “RAFT” has been used without defining it first.

Figure 1: phrases such as “insert on the upper left” and “insert on the middle” make it difficult to follow. The figure includes subfigures (a,b,...,f). Labeling the “inserts” with letters could improve the figure’s readability.

Line 159 and figure 2: Figure 2(c) shows a 3D-printed opera house, which is impressive. Line 159 reads “The integration of the RAFT agents into the 3D printing resins has been proven to yield a notable enhancement in the print resolution, facilitating the production of intricate and detailed objects with excellent resolution.” However, it is not clear how this figure and the associated discussion are useful. These shapes are not used to build the composite material. Intricately shaped nanoparticles have not been used in the next section “Photothermally-Responsive 4D Printing of LMPCs.” Overall, this part of the paper feels a bit out of place.

Figure 3b: Plots are presented as insets inside TEM photographs. Some information is lost since the photographs are overlapping the plots. Similar to previous comments about other figures, the presentation style of this figure can be improved for clarity.

Video: a typo (“laster” instead of “laser”) at 00:15 mark.

Reviewer #1

Zhang et al. grafted RAFT reagents onto liquid metal nanoparticles (RLMNPs) and successfully applied them to UV-mediated stereolithographic 3D printing and near-infrared light-responsive 4D printing. These findings highlight the potential of liquid metal-polymer composites in 4D printing and provide a promising path to developing advanced materials for soft robotics, wearables, and aerospace applications.

However, there are still some problems, such as the misunderstanding of the concept of 4D printing and the lack of highlights of this research.

Therefore, I think it is not enough to publish in Nature Communications.

We thank the reviewer for the comments and time spent evaluating the manuscript. All comments have been addressed. We also provide a more in-depth explanation of the conception of NIR-responsive 4D printing and the specific advantages highlighted in our paper.

1. The author mentioned 3D/4D printing in the abstract. However, in the manuscript, the degree of aggregation of molecular chains in the upper and lower layers of materials caused by NIR illumination is different, resulting in deformation. True 4D printing has an extra dimension of time. What is the author's understanding of 4D printing?

We appreciate the reviewer's valuable insights on the concept of 4D printing. In this paper, liquid metal nanoparticles (LMNPs) with excellent photothermal effect can raise the temperature of 3D-printed composites beyond their glass transition temperature (T_g) under NIR irradiation. T_g is the temperature at which polymers change from a hard to a more flexible state. So, this elevation in temperature triggers 3D-printed materials to recover from a bent configuration (70°) to an almost linear morphology (179°). This process differs from the deformations arising due to the varying aggregation of molecular chains in the upper and lower material layers caused by NIR illumination.

Additionally, 4D printing was initially defined as “3D printing + time” with 4D being time.¹⁻² However, the concept has also evolved in the past few years. One popular definition of 4D printing today is that the shape, property, or functionality of a 3D printed structure could be changed with time when it is exposed to a predetermined stimulus,³ such as water,⁴⁻⁵ heat,⁶⁻⁷ light,⁸⁻⁹ pH,¹⁰ and so on. In this project, we demonstrated the transformation of 3D-printed composites with time under light stimulus. The programmed shape was rapidly recovered to the original morphology within 55 seconds under NIR light irradiation, which belongs to the state-of-the-art understanding of 4D printing.

1. Q. Ge, H. J. Qi, M. L. Dunn, *Appl. Phys. Lett.* 2013, 103, 131901.
2. S. Tibbitts, *Archit. Des.* 2014, 84, 116.
3. X. Kuang, D. J. Roach, J. Wu, C. M. Hamel, Z. Ding, T. Wang, M. L. Dunn, H. J. Qi, *Adv. Funct. Mater.* 2019, 29, 1805290.
4. D. Raviv, W. Zhao, C. McKnelly, A. Papadopoulou, A. Kadambi, B. Shi, S. Hirsch, D. Dikovskiy, M. Zyacki, C. Olguin, *Sci. Rep.* 2015, 4, 7422.
5. A. S. Gladman, E. A. Matsumoto, R. G. Nuzzo, L. Mahadevan, J. A. Lewis, *Nat. Mater.* 2016, 15, 413.
6. Z. Ding, C. Yuan, X. Peng, T. Wang, H. J. Qi, M. L. Dunn, *Sci. Adv.* 2017, 3, e1602890.
7. A. Kotikian, R. L. Truby, J. W. Boley, T. J. White, J. A. Lewis, *Adv. Mater.* 2018, 30, 1706164.
8. H. Yang, W. R. Leow, T. Wang, J. Wang, J. Yu, K. He, D. Qi, C. Wan, X. Chen, *Adv. Mater.* 2017, 29, 1701627.
9. O. Kuksenok, A. C. Balazs, *Mater. Horiz.* 2016, 3, 53.

10. M. Nadgorny, Z. Xiao, C. Chen, L. A. Connal, *ACS Appl. Mater. Interfaces* 2016, 8, 28946.

Interpretations of 4D printing technologies have been added to the revised manuscript as highlighted below:

Line 35-38: “4D printing represents a cutting-edge technology in additive manufacturing, wherein initially static 3D-printed materials undergo shape transformations over time.^{1,2} 4D printing technology harnesses programmable and advanced smart materials that respond to specific stimuli, such as water,³ heat,^{4,5} photo,⁶ and pH,⁷ to achieve shape and property changes.”

Line 96-98: “These composites exhibited notable efficiency in light-triggered 4D printing, rapidly and completely restoring their pre-programmed shape when exposed to NIR light irradiation for 60 seconds.”

2. So far, many researchers have used NIR to deform liquid metal polymers. What are your advantages compared to them?

We sincerely appreciate the reviewers' thoughtful considerations regarding the innovative and advantageous aspects of our work.

In previous works, liquid metal-polymer composites for NIR-triggered deformation were developed through conventional techniques. These encompass embedding LMNPs within the polymer matrix, applying LM coatings onto the polymer matrix surface, and constructing sandwiched LM-polymer layer structures.^{1,2} In the current work, we present an innovative approach for fabricating 4D-printed LMNPs/polymer composites through the utilization of Type I Norrish photoinitiated RAFT polymerization. The advantages are outlined below:

As compared with the traditional methods, the 3D printing technique offers complex designs, rapid prototyping, and on-demand production, while 4D printing adds shape-shifting and self-assembly capabilities, enabling adaptive designs and applications in various areas.

This approach employs a well-designed RAFT agent to ensure the stabilization and uniform dispersion of LMNPs within the polymer matrix throughout the 3D printing process. This methodology, coupled with the substantiated 4D printing products, can be further utilized in the development of advanced 4D printing materials reliant on LMNPs, as well as a wide range of functional polymers.

In contrast to prior reports on liquid metal composites (with concentrations exceeding 10 vol% of LM),^{3,4} our material efficiently initiates the shape memory process using a mere 1.3 vol% of liquid metal upon exposure to near-infrared light. This advantage can be attributed to the exceptionally uniform dispersion and superior surface area of LMNPs within the polymer matrix when compared to materials produced by embedding bulk and droplet liquid metals.

1. L. Zhou, J. Ye, K. Fu, Q. Gao, Y. He, *ACS Appl. Mater. Interfaces* **2020**, *12*, 12068–12074.
2. G. G. Guymon, M. H. Malakooti, *J. Polym. Sci.* **2022**, *60*, 1300
3. X. Deng, G. Chen, Y. Liao, X. Lu, S. Hu, T. Gan, S. Handschuh-Wang, X. Zhang, *Polymers* **2022**, *14*, 2259.
4. G. Chen, X. Deng, L. Zhu, S. Handschuh-Wang, T. Gan, B. Wang, Q. Wu, H. Fang, N. Ren, X. Zhou, *J. Mater. Chem. A* **2021**, *9*, 10953-10965.

3. Why can RAFT agents be grafted onto liquid metal surfaces? FITR and UV-vis characterization methods have limitations and are not sufficient to explain. Please provide more favorable evidence.

We thank the reviewer for this question. In our project, surface grafting was accomplished using the diphosphonic acid-terminated RAFT agent, which forms a chelation bond with gallium on the LMNP surface. Our research team has previously reported on the surface anchoring effect, as follows:

1. H. Lu, S.-Y. Tang, Z. Dong, D. Liu, Y. Zhang, C. Zhang, G. Yun, Q. Zhao, K. Kalantar-Zadeh, R. Qiao, *ACS Appl. Nano Mater.* **2020**, *3*, 6905-6914.

2. X. Huang, T. Xu, A. Shen, T. P. Davis, R. Qiao, S.-Y. Tang, *ACS Appl. Nano Mater.* **2022**, *5*, 5959-5971.

Energy-dispersive X-ray spectroscopy (EDS) and X-ray photoelectron spectroscopy (XPS) were also included in the revised manuscript to elucidate the presence of phosphorus (P) originating from the RAFT agent, as well as to demonstrate the formation of the Ga-O-P bonds, as below:

Line 118-119: The P signal of RAFT agents was also detected in Energy-dispersive X-ray spectroscopy of RLMNPs (Figure S2).

Figure S2. EDS analysis of RLMNPs including Ga, In, O, and P elements.

Line 125-127: X-ray photoelectron spectroscopy (XPS) analyses demonstrate the presence of P signals originating from RLMNPs, as indicated in Figure S3.

Figure S3. The XPS spectrum of LMNPs and RLMNPs. (a) a typical XPS spectrum, and a high-resolution XPS of (b) Ga 2p_{3/2} and (c) P 2p.

4. The author said that the resolution of 3D printing has been improved, how much has the resolution been improved? Detailed proof is needed that the resolution has been increased.

We thank the reviewer for this question. In response, we carried out additional experiments involving the printing of square, disc, and triangular objects to assess the enhancement in resolution, and details are provided in the revised manuscript below:

Line 159-165: “One of the advantages of integrating RAFT agents into 3D printing is to improve the resolution and accuracy of fabricated materials.³⁹ Under a consistent layer cure time of 40 s, objects with satisfactory resolution were successfully fabricated (Figure S5a). Conversely, regions adjacent to RAFT-free 3D-printed objects exhibited premature curing in identical conditions, resulting in the formation of polymerized masses and consequent degradation of object resolution (Figure S5b). RAFT agents would be regarded as a light-absorbing dye, effectively mitigating light scattering and thereby enhancing print resolution across all spatial axes.³⁹”

A figure was added to the supporting information:

Figure S5. (a) The image of 3D-printed objects with RAFT agents using a layer cure time of 40 s; (b) The image of 3D-printed objects without RAFT agents under an identical condition.

5. The resolution of SEM and mapping images is too low.

We greatly appreciate the reviewer's feedback. In response, updated SEM and elemental mapping images with improved resolution are provided in Figure 2b of the revised manuscript.

6. The authors mention that one-step printing of liquid metal nanoparticle-based resins can be applied to other 3D applications, such as bioprinting. However, the authors did not explore its biocompatibility. TPO and PEGDA are both toxic substances, why can they be used in bioprinting?

We are grateful to the reviewer for raising this important question. In response, we undertook a cell viability study by using various cell lines, including human embryonic kidney cells (HEK-293) and mouse macrophage cells (Raw 264.7). The results of this study revealed no significant toxicity

associated with the utilization of our 3D-printed materials, thus indicating a high level of biocompatibility. (Figure S14)

Despite the potential toxicity of TPO, their substantial entrapment within the polymer matrix appears to serve as a safeguard against inducing cell toxicity. Most importantly, we anticipate the low toxicity of our material was attributed to the minimal amount of TPO (<0.3%). In addition, PEGDA-based materials have been reported to be non-toxic.¹ The polymerization of PEGDA has also significantly reduced the toxicity arising from the double bond of monomers.

1. K. McAvoy, D. Jones, R. R. S. Thakur, *Pharm. Res.* 2018, 35, 1-17

The following information was added to the manuscript and supporting information:

Results and Discussion:

Line 253-256: “We further evaluated the cytotoxicity of the 3D-printed materials (Figure S14) on *in vitro* cell models using human embryonic kidney cells (HEK-293) and mouse macrophage cells (Raw 264.7). No significant toxicity was observed on both cell lines, indicating excellent biocompatibility of our 3D-printed materials.

Line 474-477: Methods: **Biotoxicity test of 3D-printed materials**

HEK-293 and Raw 264.7 were seeded in 24 well plates and co-cultured with 3D printed materials for 24 h. Afterward, 50 μ L AlamarBlue reagent was added to each well. Absorbance was measured at 570/600 nm after 3 h incubation using the EnSight plate reader.

Figure S14. Cell viability studies with and without 3D-printed objects on RAW-264.7 and HEK-293 cells.

7. The authors mention that intramolecular hydrogen bonding through amide groups leads to increased flexibility and stretchability. Why are intramolecular hydrogen bonds produced instead of intermolecular hydrogen bonds? And confusingly, can hydrogen bonding enhance both tensile strength and flexibility?

To clarify, HEAAm-based materials should have both intramolecular and intermolecular hydrogen bonds due to the presence of amide and hydroxyl groups. The prerequisite for the intramolecular hydrogen bonds is the coexistence of donor and acceptor atoms within the same molecule and enough steric flexibility. The rest three monomers could only generate intermolecular hydrogen bonds.¹⁻³ Furthermore, hydrogen bonds are constantly destroyed and formed in the process of tensile, which makes the material present viscoelasticity and improves its tensile strength.⁴

1. Filarowski, *J. Phys. Org. Chem.* 2005, 18, 686-698.
2. P. I. Nagy, *Int. J. Mol. Sci.* 2014, 15, 19562-19633
3. K. Józwiak, A. Jezierska, J. J. Panek, E. A. Goremychkin, P. M. Tolstoy, I. G. Shenderovich, A. Filarowski, *Molecules* 2020, 25, 4720
4. M. Wang, S. Liang, W. Gao, Y. Qin, *R. Soc. Open Sci.* 2022, 9, 211393.

The following discussion was added to the revised manuscript:

Line 244-250: For example, HEAAm-based objects demonstrated superior mechanical properties, with the highest tensile stress (47.81 ± 2.52 MPa) and storage modulus (4170.18 ± 26.00 MPa) compared to materials generated by the other three monomers. This phenomenon can be attributed to amide and hydroxyl groups in HEAAm, which facilitate the formation of both intramolecular and intermolecular hydrogen bonds.⁵¹⁻⁵³ These hydrogen bonds can improve the flexibility and tensile strength of 3D-printed materials by undergoing continuous breaking and reforming during the tensile process.⁵⁴

8. Deformation caused by NIR is not novel, and only 10 reciprocating recoveries are not sufficient to demonstrate stable recovery.

The innovation of our work primarily centers around the development of a versatile 3D printing technique using the type I photoinitiator-RAFT polymerization, which enables the fabrication of LMNPs/polymer composites with NIR-responsive 4D printing capabilities.

Furthermore, in the revised manuscript, the 3D-printed LMNPs/polymer composites underwent up to 25 reciprocating recovery cycles. There were no significant changes observed in terms of recovery efficiency, indicating their excellent recovery stability. The number of cycles achieved using our material surpasses those reported for similar materials,¹ and meets the criteria for commercial SMPs that are examined through a series of at least 20 thermo-mechanical cycles.²

1. Y. Y. C. Choong, S. Maleksaedi, H. Eng, J. Wei, P.-C. Su, *Materials & Design* **2017**, 126, 219-225.
2. S. C. Arzberger, M. L. Tupper, M. S. Lake, R. Barrett, K. Mallick, C. Hazelton, W. Francis, P. N. Keller, D. Campbell, S. Feucht, in *Smart structures and materials 2005: industrial and commercial applications of smart structures technologies*, Vol. 5762, SPIE, 2005, pp. 35-47.

Figures and discussion were added as below:

Figure 5. Potential applications of NIR-responsive 4D printing with 1 wt% RLMNPs. (a) The recycling process of shape memory polymers in repeated programming and photoinduced 4D printing; (b) The recovery angle on time of LMPCs in a total of 25 cycles while irradiating with 808 nm laser (0.3 W/cm^2) for 60 s.

Figure S20. The SEM photograph of the deformed part of 3D-printed LMPCs after performing shape memory cycles 25 times.

Figure S21. Tensile tests of 3D printed objects in a total of 25 cycles.

Discussion:

Line 338-342: Importantly, the recovery angles of LMPCs remained unaffected through a minimum of 25 cycles of programming and NIR light irradiation as depicted in Figure 5b, demonstrating their robust shape memory recyclability. SEM photographs confirmed the stable nanostructures of RLMNPs in 3D-printed materials (Figure S20) and mechanical tests indicated similar tensile strength in 25 cycles (Figure S21).

9. What are the specific applications of RLMNP-based materials? I think only the recovery of deformation and weightlifting is not enough to prove its application, this is only performance.

We appreciate the question raised by the reviewer. In response, we have conducted additional experiments to demonstrate the diverse potential applications of RLMNP-based materials across various scenarios. These applications encompass facilitating object release, enabling object grasping and releasing functionalities, as well as effectively controlling rotational bistable structures.

The following discussion and Figure 6 were added to the revised manuscript:

Line 361-400: **Application of the 3D-printed LMPCs as NIR-Responsive Soft Robots**

Harnessing the rapid photothermal responsiveness of LMPCs accompanied by a large deformation capacity and robust force generation, we explored the use of LMPC for the development of soft robotic systems. As shown in Figure 6a, we designed and printed a flower petal-like soft robot comprised of LMPCs, which were subsequently programmed to a temporary shape capable of grasping a cap. Upon NIR irradiation ($\lambda_{\max} = 808 \text{ nm}$, 0.3 w/cm^2) at the top of the robot, it rapidly returned to its pre-deformation state, analogous to the unfurling of flower petals, resulting in the release of the cap (Figure 6b). Nevertheless, in the absence of irradiation, the programmed soft robot consistently maintained its grip on the object (Video S2).

In another scenario, object grasping and release can be achieved by directing irradiation to specific positions on the 3D-printed soft robots (Figure 6c). Following a 40 s exposure to NIR light at the base region of the programmed device, a rapid transition from a curved to a straightened state occurred. This deformation caused the bottom grippers to close, successfully capturing the cap. Subsequently, by adjusting the NIR light to irradiate the upper region of the device, a gradual unfolding of the entire material was observed, resulting in the release of the held object (Figure 6d and Video S3).

Additionally, LMPCs' shape memory capability, coupled with the 3D printing technology developed in this study, empowers the creation of bistable structures tailored for locomotion in soft robotics. In this case, rotational bistable components were connected by two 3D-printed tuning arms comprised of LMPCs ($1 \times w \times h = 10 \times 2 \times 1 \text{ mm}$). As shown in Figure 6e, the inner part can be inter-rotated into two distinct stable states (A and B), while the outer ring remains fixed in place. At room temperature, the tuning arm is curved due to the programmed LMPCs, and the inner part is rotated 21° to the left, maintaining stable State B. Upon 30 s of NIR irritation, the deformed tuning arms returned to their pre-deformation state, and the inner part shifted 21° to the right, transforming into stable State A (Figure 6f). Consequently, by employing LMPCs elements, we demonstrate the feasibility of tunable bistable structures as soft actuators for mechanical switches, showcasing promising applications in the medical device sector, consumer electronics, and the automotive industry.

Figure 6. Application of the 3D-printed LMPCs as NIR-responsive soft robots. (a) The schematic diagram and (b) demonstration of releasing an object via NIR light responsive LMPCs. The scale bar corresponds to 10 mm (c) The schematic diagram and (d) demonstration of gripping and releasing an object via using NIR light controlled LMPCs. The scale bar corresponds to 10 mm (e) The schematic diagram and (f) demonstration of the rotated structure (state B) transforms into the original shape (state A) upon NIR exposure. The dotted gray arrow shows the program direction, and the dotted red arrow shows the rotation direction. The scale bar corresponds to 5 mm.

Reviewer #2

This is an interesting and exciting new study showing a new method to create 4D printed structures that could be used as soft robots in the NIR region.

The study is very well done and has merit for publication. I have just a few minor remarks that would be needed to tackle before the manuscript is ready for publication:

We thank the reviewer for the positive comment and time spent on evaluating the manuscript. All reviewer comments have been adequately addressed.

1. It is not clear why the use of nanoparticles in other approaches would be a drawback in terms of altered mechanical properties or physical characteristics of the final composites. One could certainly modulate such properties to an extent, therefore being more of an advantage than a disadvantage. This is discussed in the introduction but should be better elaborated.

We express our gratitude to the reviewer for raising this question. In response, we have revised the introduction and expanded upon the advantages and disadvantages of incorporating rigid nanoparticles in 4D printing.

Line 43-65: “To overcome the limitations of using polymers alone, several researchers have demonstrated the integration of functional rigid nanomaterials into 3D-printed presents several benefits for advancing the 4D printing nanocomposites.¹⁴ Nanoparticles with unique properties (such as photosensitivity and chemosensitivity) can enhance the stimuli-responsiveness of printed objects, thereby enabling precise and efficient shape changes. For instance, Kuhnt et al. successfully integrated magnetic iron oxide nanoparticles into 3D-printed objects, thereby producing composites that demonstrated thermally and remotely controlled shape-memory behavior under an alternating magnetic field.¹⁵ An alternative approach to producing composites capable of inducing shape transformation involved the incorporation of conductive carbon black particles or carbon nanotubes into 3D-printed polymers, thus sensitizing the composites to electrothermal stimulation. Moreover, rigid nanomaterials can also reinforce the mechanical properties of products, elevating overall durability and augmenting the load-bearing capacity of the composite material for diverse applications.

Nevertheless, a drawback to using particulate-based nanocomposites to achieve shape change lies in establishing the necessary percolated network within the elastomer matrix, often requiring high filler loadings. These elevated loadings complicate the printing process and limit the extent of shape change.^{16,17} Furthermore, the utilization of rigid nanoparticles may compromise the softness and flexibility of composites, thereby limiting the shape-morphing capabilities of 4D printed materials within intricate environments. In addition, some studies have reported that integrating rigid nanoparticles can increase the glass transition temperature (T_g) and melting temperature (T_m) of polymers in thermal-responsive 4D printing,^{18,19} leading to an increase in the temperature required for shape recovery, and constraining the spectrum of stimuli capable of inducing deformation.”

2. I miss a final soft robotic application. In which application would the proposed liquid metal-polymer composites be of use? The experiments performed beyond the more fundamental characterization of the new system aim at creating demonstrations more than real applications.

We thank the reviewer for this suggestion. In response, we have conducted additional experiments to demonstrate the diverse potential applications of RLMNP-based materials across various scenarios. These applications encompass facilitating object release, enabling object grasping and releasing functionalities, as well as effectively controlling rotational bistable structures.

The following discussion and Figure 6 were added to the revised manuscript:

Line 361-400: **Application of the 3D-printed LMPCs as NIR-Responsive Soft Robots**

Harnessing the rapid photothermal responsiveness of LMPCs accompanied by a large deformation capacity and robust force generation, we explored the use of LMPC for the development of soft robotic systems. As shown in Figure 6a, we designed and printed a flower petal-like soft robot comprised of LMPCs, which were subsequently programmed to a temporary shape capable of grasping a cap. Upon NIR irradiation ($\lambda_{\max} = 808 \text{ nm}$, 0.3 w/cm^2) at the top of the robot, it rapidly returned to its pre-deformation state, analogous to the unfurling of flower petals, resulting in the release of the cap (Figure 6b). Nevertheless, in the absence of irradiation, the programmed soft robot consistently maintained its grip on the object (Video S2).

In another scenario, object grasping and release can be achieved by directing irradiation to specific positions on the 3D-printed soft robots (Figure 6c). Following a 40 s exposure to NIR light at the base region of the programmed device, a rapid transition from a curved to a straightened state occurred. This deformation caused the bottom grippers to close, successfully capturing the cap. Subsequently, by adjusting the NIR light to irradiate the upper region of the device, a gradual unfolding of the entire material was observed, resulting in the release of the held object (Figure 6d and Video S3).

Additionally, LMPCs' shape memory capability, coupled with the 3D printing technology developed in this study, empowers the creation of bistable structures tailored for locomotion in soft robotics. In this case, rotational bistable components were connected by two 3D-printed tuning arms comprised of LMPCs ($1 \times w \times h = 10 \times 2 \times 1 \text{ mm}$). As shown in Figure 6e, the inner part can be inter-rotated into two distinct stable states (A and B), while the outer ring remains fixed in place. At room temperature, the tuning arm is curved due to the programmed LMPCs, and the inner part is rotated 21° to the left, maintaining stable State B. Upon 30 s of NIR irritation, the deformed tuning arms returned to their pre-deformation state, and the inner part shifted 21° to the right, transforming into stable State A (Figure 6f). Consequently, by employing LMPCs elements, we demonstrate the feasibility of tunable bistable structures as soft actuators for mechanical switches, showcasing promising applications in the medical device sector, consumer electronics, and the automotive industry.

Figure 6. Application of the 3D-printed LMPCs as NIR-responsive soft robots. (a) The schematic diagram and (b) demonstration of releasing an object via NIR light responsive LMPCs. The scale bar corresponds to 10 mm (c) The schematic diagram and (d) demonstration of gripping and releasing an object via using NIR light controlled LMPCs. The scale bar corresponds to 10 mm (e) The schematic diagram and (f) demonstration of the rotated structure (state B) transforms into the original shape (state A) upon NIR exposure. The dotted gray arrow shows the program direction, and the dotted red arrow shows the rotation direction. The scale bar corresponds to 5 mm.

Reviewer #3

The manuscript presents a new advanced material, a composite of soft and liquid nanoparticles and polymer, that can change shapes once by NIR irradiation. Instead of rigid nanoparticles that are conventionally used in such composites, soft and liquid particles are used. This gets rid of the undesired effects arising from the rigidity of the nanoparticles. The authors report that the soft inclusions reduce glass transition temperature, tensile stress, and modulus of the composite. These observations are opposite to what happens in the case of rigid nanoparticles.

The manuscript is well written, and the results can have significant implications in the field of soft robotics. However, it requires a major revision before publication in Nature Communications. In summary, a significant gap exists between the material manufacturing and its application in “soft robots” (a word that is in the title of the manuscript). The paper reads like a traditional materials science paper focusing on manufacturing and characterization without demonstrating it in soft robots. Detailed comments follow.

We thank the reviewer for the comments and time spent on evaluating the manuscript. All reviewer comments have been adequately addressed.

1. Quantitative and direct comparisons between the proposed composite (that uses liquid nanoparticles) and comparable composites with rigid particles should be provided. Otherwise, the reader is left wondering what’s the point of using liquid and soft particles?

We thank the reviewer for the invaluable suggestion. In response, we have conducted further experiments to create similar composites by incorporating iron oxide nanoparticles (IONPs) into the 3D printing process. We have subsequently conducted a thorough characterization of these composites, focusing on their photothermal efficiency and 4D printing capabilities. The comparison between the composites based on liquid metal nanoparticles and those utilizing rigid iron oxide nanoparticles is presented in the revised manuscript, as outlined below.

Figure S18. (a) 0.3 W/cm² laser heating 3D-printed materials that contained RLMNPs and iron oxide nanoparticles (IONPs), respectively, ambient temperature: 25 °C; (b) Recovery curves on time of 3D-

printed materials that respectively containing RLMNPs and IONPs while irradiating with 808 nm laser (0.3 W/cm^2) for 60 s.

Line 323-330: “Subsequently, we conducted a comparative analysis to evaluate the 4D performance of composites prepared by using rigid and liquid nanoparticles, respectively. In this context, iron oxide nanoparticles (IONPs) were selected for their similar photothermal conversion efficiency to LMNPs.⁵⁸ We observed that both types of composites exhibited a comparable temperature increase under laser irradiation for 180 s ($\lambda_{\text{max}} = 808 \text{ nm}$, 0.3 W/cm^2 , Figure S18a). However, the LMPCs outperformed the IONP-based composites in terms of shape recovery capacity, achieving a recovery angle of 99% compared to 91% in IONP-based composites (Figure S18b), which can be attributed to the softness of LMNPs in contrast to the rigidity of IONPs.”

2. A major criticism is the potential application of the composite to build soft robots. Only two shapes (a bent bar and a coiled spring-like object) are shown. The shape change is rather slow. For publication in a top generalist journal like Nature Communications, I would expect the paper to include more practical demonstrations. Even if the authors cannot go all the way to building soft robots, they should be able to show some simple demonstrations like grasping a light object (say, a pen) and releasing it once exposed to NIR.

We thank the reviewer for this suggestion. We have incorporated soft robot-related applications of composites into the manuscript. These include facilitating object release, enabling object grasping/releasing functionalities, and effectively controlling rotational bistable structures.

The following discussion and figure 6 were added to the manuscript:

Line 361-400: **Application of the 3D-printed LMPCs as NIR-Responsive Soft Robots**

Harnessing the rapid photothermal responsiveness of LMPCs accompanied by a large deformation capacity and robust force generation, we explored the use of LMPC for the development of soft robotic systems. As shown in Figure 6a, we designed and printed a flower petal-like soft robot comprised of LMPCs, which were subsequently programmed to a temporary shape capable of grasping a cap. Upon NIR irradiation ($\lambda_{\text{max}} = 808 \text{ nm}$, 0.3 w/cm^2) at the top of the robot, it rapidly returned to its pre-deformation state, analogous to the unfurling of flower petals, resulting in the release of the cap (Figure 6b). Nevertheless, in the absence of irradiation, the programmed soft robot consistently maintained its grip on the object (Video S2).

In another scenario, object grasping and release can be achieved by directing irradiation to specific positions on the 3D-printed soft robots (Figure 6c). Following a 40 s exposure to NIR light at the base region of the programmed device, a rapid transition from a curved to a straightened state occurred. This deformation caused the bottom grippers to close, successfully capturing the cap. Subsequently, by adjusting the NIR light to irradiate the upper region of the device, a gradual unfolding of the entire material was observed, resulting in the release of the held object (Figure 6d and Video S3).

Additionally, LMPCs' shape memory capability, coupled with the 3D printing technology developed in this study, empowers the creation of bistable structures tailored for locomotion in soft robotics. In this case, rotational bistable components were connected by two 3D-printed tuning arms comprised of LMPCs ($1 \times w \times h = 10 \times 2 \times 1 \text{ mm}$). As shown in Figure 6e, the inner part can be inter-rotated into two

distinct stable states (A and B), while the outer ring remains fixed in place. At room temperature, the tuning arm is curved due to the programmed LMPCs, and the inner part is rotated 21° to the left, maintaining stable State B. Upon 30 s of NIR irradiation, the deformed tuning arms returned to their pre-deformation state, and the inner part shifted 21° to the right, transforming into stable State A (Figure 6f). Consequently, by employing LMPCs elements, we demonstrate the feasibility of tunable bistable structures as soft actuators for mechanical switches, showcasing promising applications in the medical device sector, consumer electronics, and the automotive industry.

Figure 6. Application of the 3D-printed LMPCs as NIR-responsive soft robots. (a) The schematic diagram and (b) demonstration of releasing an object via NIR light responsive LMPCs. The scale bar corresponds to 10 mm (c) The schematic diagram and (d) demonstration of gripping and releasing an object via using NIR light controlled LMPCs. The scale bar corresponds to 10 mm (e) The schematic diagram and (f) demonstration of the rotated structure (state B) transforms into the original shape (state A) upon NIR exposure. The dotted gray arrow shows the program direction, and the dotted red arrow shows the rotation direction. The scale bar corresponds to 5 mm.

3. Longer cyclic loading should be investigated. Fig. 4 (g) includes data on material properties after one cycle of loading. But what is the limit? When does the material stop functioning (i.e., loses its ability to recover original shapes even with NIR irradiation)?

We conducted additional cyclic loading tests of LMPCs to demonstrate their performance in longer cyclic loading. Our results indicate that there were no significant changes in terms of tensile stress and Young's modulus as shown in Figure S21. The number of cycles achieved with our material surpasses those reported for similar materials.¹

However, when repeated 30 times, shape recovery efficiency dropped notably at the same irradiation time, alongside a 27% reduction in the tensile strength of the original object (Figure S22). Because of limitations in our instruments, it's not currently possible to perform numerous cycle tests (in the thousands or hundreds) until the materials completely stop functioning.

1. Y. Y. C. Choong, S. Maleksaeedi, H. Eng, J. Wei, P.-C. Su, *Materials & Design* **2017**, *126*, 219-225.

Figures and the discussion were added in the manuscript:

Figure S21. Tensile tests of 3D printed objects in a total of 25 cycles.

Figure S22. (a) Recovery Angle of LMPCs at the first and 30th cycle of 4D printing when irradiation with NIR light for 60 seconds; (b) Tensile stress, and (c) Young's modulus of LMPCs at the first and 30th cycle of 4D printing.

Line 340-344: SEM photographs confirmed the stable nanostructures of RLMNPs in 3D-printed materials (Figure S20) and mechanical tests indicated similar tensile strength over 25 cycles (Figure

S21). When repeated 30 times, recovery efficiency notably dropped at the same irradiation time, alongside concurrent reductions of 25 % in tensile stress and 20% in Young's modulus of the original object (Figure S22).

4. The title could be improved, or the manuscript could be augmented so that the title is indeed a summary of the paper. The title gives the sense that a “3D-printed liquid metal polymer composite,” which is a type of material, is equivalent to “NIR-response 4D printing soft robot, which is a manufacturing technique. As mentioned earlier, not much is discussed about the application of the material in soft robots.

Thanks for the suggestion. In response, Figure 6 and discussion related to applications of soft robotics have been added to the manuscript to enhance its alignment with the title.

5. Line 20: The abbreviation “RAFT” has been used without defining it first.

The abbreviation has been revised in the manuscript. We have carefully verified all abbreviations to ensure they are defined before being used.

Line 22-25: “In this study, reversible addition-fragmentation chain transfer polymerization (RAFT) agents grafted onto liquid metal nanoparticles (RLMNPs) have been successfully employed in ultraviolet light-mediated stereolithographic 3D printing and near-infrared light-responsive 4D printing.”

6. Figure 1: Phrases such as “insert on the upper left” and “insert on the middle” make it difficult to follow. The figure includes subfigures (a,b,...,f). Labeling the “inserts” with letters could improve the figure's readability.

We have revised related Figure 1 according to your suggestion to improve the readability.

Figure 1. The preparation and characterization of RLMNPs. (a) The schematic diagram of the fabrication of RLMNPs in an ethanol solution via ultrasonication; (b) Photograph and the TEM image of RLMNPs; (c) Intensity-based size distribution histograms of RLMNPs measured by DLS; (d) EDS analysis of RLMNPs, including Ga, In, O, P and S elements; (e) Zeta potential, (f) UV-vis spectra, and (g) FTIR spectra of LMNPs, RAFT agents and RLMNPs.

7. Line 159 and figure 2: Figure 2(c) shows a 3D-printed opera house, which is impressive. Line 159 reads “The integration of the RAFT agents into the 3D printing resins has been proven to yield a notable enhancement in the print resolution, facilitating the production of intricate and detailed objects with excellent resolution.” However, it is not clear how this figure and the associated discussion are useful. These shapes are not used to build the composite material. Intricately shaped nanoparticles have not been used in the next section “Photothermally-Responsive 4D Printing of LMPCs.” Overall, this part of the paper feels a bit out of place.

We apologize for any confusion elicited in our previous description. The following discussion and figure were added in the manuscript to make it clearer:

Line 159-170: “One of the advantages of integrating RAFT agents into 3D printing is to improve the resolution and accuracy of fabricated materials.³⁹ Under a consistent layer cure time of 40 s, objects with satisfactory resolution were successfully fabricated (Figure S5a). Conversely, regions adjacent to RAFT-free 3D-printed objects exhibited premature curing in identical conditions, resulting in the formation of polymerized masses and consequent degradation of object resolution (Figure S5b). RAFT agents would be regarded as a light-absorbing dye, effectively mitigating light scattering and thereby enhancing print resolution across all spatial axes.³⁹ Additionally, we employed RLMNPs to fabricate intricate designs resembling the iconic Sydney Opera House (Figure 2c) and snowflakes (Figure S6) with exceptional structural fidelity. This highlights the benefits of incorporating RAFT agents into 3D

printing, particularly in addressing the exacting resolution requirements of sophisticated structures in 4D-printed soft robots.”

(a) 3D-printed objects with RAFT agents

(b) 3D-printed objects without RAFT agents

Figure S5. (a) The image of 3D-printed objects with RAFT agents using a layer cure time of 40 s; (b) The image of 3D-printed objects without RAFT agents under an identical condition.

8. Figure 3b: Plots are presented as insets inside TEM photographs. Some information is lost since the photographs are overlapping the plots. Similar to previous comments about other figures, the presentation style of this figure can be improved for clarity.

Following your suggestion, the TEM photographs and DLS data have been segregated into two distinct figures as shown in Figure 3.

Figure 3. (b) TEM photographs of RLMNPs in liquid resins with different monomers; (c) Intensity-based size distribution histograms of RLMNPs in liquid resins with different monomers measured by DLS. The size of RLMNPs was measured immediately after sonication.

9. Video: a typo (“laster” instead of “laser”) at 00:15 mark.

Laster has been changed to laser. Please see the modified video.

REVIEWER COMMENTS

Reviewer #1 (Remarks to the Author):

Thank you very much for the author's prompt response. However, I still feel that there are some issues that the author has not explained clearly. Specifically, as follows:

1. Regarding the fourth question about resolution, it cannot be explained solely through macroscopic appearance; microscopic characterization should be provided.
2. In the fifth question, the author's SEM mapping shows no phosphorus (P) element, but XPS and EDS show the presence of phosphorus (P) element. Please ask the authors to provide an explanation.
3. In the seventh question, the authors mentioned that hydrogen bonds can enhance the material's mechanical properties. I am aware that hydrogen bonds can improve mechanical properties. However, hydrogen bonds are weak bonds, and it seems unlikely that the material's mechanical properties would improve so significantly solely through the action of hydrogen bonds. Please ask the authors to provide a more in-depth explanation of the strengthening mechanism."

Reviewer #2 (Remarks to the Author):

The revised paper has significantly improved, clarifying over the several missing points that were raised by the reviewers.

I am pleased with this version and think it is now ready for publication.

Reviewer #3 (Remarks to the Author):

The authors made substantial improvements to the manuscript. The demonstrations in soft robotic applications (Fig. 6 of the main manuscript) are commendable. However, the proposed material cannot be used for more than 30 cycles. Quoting the authors, "when repeated 30 times, shape recovery efficiency dropped notably at the same irradiation time, alongside a 27% reduction in the tensile strength of the original object." This leaves us wondering whether the proposed material can truly be useful in soft robotics.

Reviewer #1

Thank you very much for the author's prompt response. However, I still feel that there are some issues that the author has not explained clearly. Specifically, as follows:

We thank the reviewer for the valuable comments and time dedicated to evaluating the manuscript.

1. Regarding the fourth question about resolution, it cannot be explained solely through macroscopic appearance; microscopic characterization should be provided.

We appreciate the feedback from the reviewer. In response, we used SEM to visualize and assess the resolution of a 3D-printed porous structure with and without the incorporation of RAFT agents. Detailed information is provided in the revised manuscript and accompanying figures can be found in the supporting document, as below:

Line 166-169: Microscopic characterization was employed to further evaluate the resolution of 3D-printed objects. The SEM image showcased clear pore structures of the 3D-printed porous object with RAFT (Figure S5c). Notably, these hollows were absent in the RAFT-free 3D-printed object (Figure S5d).

A figure was updated in the supporting information:

Figure S5. Resolution comparison of 3D-printed objects with and without RAFT agents (a) Photo of 3D-printed objects with RAFT agents using a layer cure time of 40 s; (b) Photo of the RAFT-free 3D-printed object under an identical condition; (c) SEM image of the 3D-printed porous object with RAFT agents. The insert on the upper left displays a photo of the 3D-printed porous object with RAFT agents; (d) SEM image of the RAFT-free 3D-printed

porous object. The insert on the upper left displays a photo of the RAFT-free 3D-printed porous object.

2. In the fifth question, the author's SEM mapping shows no phosphorus (P) element, but XPS and EDS show the presence of phosphorus (P) element. Please ask the authors to provide an explanation.

We thank the reviewer for this question. We apologize for the missing of the P element in SEM mapping. Upon closer examination, we did indeed observe the presence of the phosphorus (P) element, which has now been included in Figure 2b of the revised manuscript.

Figure 2. 3D printing and characterizations of LMPCs. (a) The fabrication process of 3D-printed objects by using TPO as photo-initiator, TBAm as the monomer, PEGDA as a crosslinker, and RLMNPs as RAFT agents; (b) SEM photograph and EDS elemental mapping of 3D-printed objects with RLMNPs; the detected elements included Ga, In, P and S; (c) SEM photograph and EDS elemental mapping of RLMNPs in 3D-printed LMPCs; (d) A model resembling the Sydney Opera House (1 wt% of RLMNPs) was manufactured using stereolithography 3D printing; (e) Mechanical properties of LMPCs and LMPCs without RAFT agents; (f) Mechanical properties of LMPCs containing different concentrations of RLMNPs (0, 0.5, 1, and 2 wt%). Data are presented as mean values \pm SD; n = 3. Statistical significance is calculated based on Welch's t-test (e) and one-way ANOVA (f). *P < 0.05, **P < 0.01, and ***P < 0.001. Source data are provided as a Source Date file.

Line 154-158: Scanning electron microscopy (SEM) and elemental mapping observed a smooth surface without any defects and the presence of essential elements (Ga, In, P and S) in the 3D-printed materials (Figure 2b). In contrast, the S and P elements could not be detected in 3D-printed LMNP-polymer composites without RAFT agents due to the lack of RAFT agents (Figure S4).

3. In the seventh question, the authors mentioned that hydrogen bonds can enhance the material's mechanical properties. I am aware that hydrogen bonds can improve mechanical properties. However, hydrogen bonds are weak bonds, and it seems unlikely that the material's mechanical properties would improve so significantly solely through the action of hydrogen bonds. Please ask the authors to provide a more in-depth explanation of the strengthening mechanism."

While hydrogen bonds are indeed considered weak compared to covalent or ionic bonds, hydrogen-bonding interactions have been widely reported to show remarkably high mechanical strength and elasticity of polymer complexes.¹⁻⁴ This is primarily due to the high abundance of hydrogen bonding sites in materials, which allows for a collective and synergistic effect to increased strength, stiffness, and other mechanical properties.

1. L. A. Galuska, M. U. Ocheje, Z. C. Ahmad, S. Rondeau-Gagné, X. Gu, *Chem Mater* 2022, 34, 2259-2267.
2. R. Nasser, N. Bouzari, J. Huang, H. Golzar, S. Jankhani, X. Tang, T. H. Mekonnen, A. Aghakhani, H. Shahsavan, *Nature Communications* 2023, 14, 6108;
3. W. Wang, H. Liu, L. Pei, H. Liu, M. Wang, S. Li, Z. Wang, *Eur Polym J* 2023, 182, 111717;
4. Y. Wang, X. Liu, S. Li, T. Li, Y. Song, Z. Li, W. Zhang, J. Sun, *ACS applied materials & interfaces* 2017, 9, 29120-29129.

Line 254-260: This phenomenon can be attributed to amide and hydroxyl groups in HEAAm, which facilitate the formation of numerous intramolecular and intermolecular hydrogen bonds.⁵¹⁻⁵³ The substantial number of hydrogen bonds collectively and synergistically

enhances the flexibility and tensile strength of 3D-printed materials.⁵⁴ Moreover, these hydrogen bonds continually undergo a process of breaking and reforming during tensile forces, further contributing to the material's mechanical properties.⁵⁵

The author has already provided a satisfactory response to my questions and demonstrates the ability to publish in Nature Communications. I recommend acceptance.